# Multi-year El Niño events tied to the North Pacific Oscillation

Ruiqiang Ding [1✉], Yu-Heng Tseng [2], Emanuele Di Lorenzo [3], Liang Shi[4], Jianping Li [5✉], Jin-Yi Yu [6], Chunzai Wang [7], Cheng Sun [8], Jing-Jia Luo[9], Kyung-Ja Ha [10], Zeng-Zhen Hu [11] & Feifei Li[12]

Multi-year El Niño events induce severe and persistent floods and droughts worldwide, with significant socioeconomic impacts, but the causes of their long-lasting behaviors are still not fully understood. Here we present a two-way feedback mechanism between the tropics and extratropics to argue that extratropical atmospheric variability associated with the North Pacific Oscillation (NPO) is a key source of multi-year El Niño events. The NPO during boreal winter can trigger a Central Pacific El Niño during the subsequent winter, which excites atmospheric teleconnections to the extratropics that re-energize the NPO variability, then re-triggers another El Niño event in the following winter, finally resulting in persistent El Niño-like states. Model experiments, with the NPO forcing assimilated to constrain atmospheric circulation, reproduce the observed connection between NPO forcing and the occurrence of multi-year El Niño events. Future projections of Coupled Model Intercomparison Project phases 5 and 6 models demonstrate that with enhanced NPO variability under future anthropogenic forcing, more frequent multi-year El Niño events should be expected. We conclude that properly accounting for the effects of the NPO on the evolution of El Niño events may improve multi-year El Niño prediction and projection.

[1] State Key Laboratory of Earth Surface Processes and Resource Ecology, Beijing Normal University, Beijing, China. [2] Institute of Oceanography, National Taiwan University, Taipei, Taiwan. [3] School of Earth and Atmospheric Sciences, Georgia Institute of Technology, Atlanta, GA 30332, USA. [4] Key Laboratory for Semi-Arid Climate Change of the Ministry of Education, College of Atmospheric Sciences, Lanzhou University, Lanzhou, China. [5] Frontiers Science Center for Deep Ocean Multispheres and Earth System (FDOMES)/Key Laboratory of Physical Oceanography/Institute for Advanced Ocean Studies, Ocean University of China, Qingdao, China. [6] Department of Earth System Science, University of California, Irvine, CA 92697, USA. [7] State Key Laboratory of Tropical Oceanography, South China Sea Institute of Oceanology, Chinese Academy of Sciences, Guangzhou 510301, China. [8] College of Global Change and Earth System Science (GCESS), Beijing Normal University, Beijing, China. [9] Institute for Climate and Application Research (ICAR), Nanjing University of Information Science and Technology, Nanjing 210044, China. [10] Center for Climate Physics, Institute for Basic Science, and Department of Atmospheric Sciences, Pusan National University, Busan 609-735, Korea. [11] Climate Prediction Center, NCEP/NWS/NOAA, College Park, MD, USA. [12] State Key Laboratory of Numerical Modeling for Atmospheric Sciences and Geophysical Fluid Dynamics (LASG), Institute of Atmospheric Physics, Chinese Academy of Sciences, Beijing, China. ✉email: drq@bnu.edu.cn; ljp@ouc.edu.cn

The El Niño/Southern Oscillation (ENSO) is the dominant climate phenomenon in the tropical ocean affecting the global climate and extreme weather conditions[1–5]. Typically, El Niño and La Niña episodes develop during the boreal summer, peak during early winter, and decay rapidly during the following spring, lasting about 9–12 months. However, not all ENSO events are the same[6]. Some La Niña events persist through the following year and often re-intensify in the subsequent winter, lasting two years or longer[7]. Although less frequent than multi-year La Niña events, multi-year El Niño events are also occasionally observed in the tropical Pacific (such as the 2014/15/16 El Niño event)[7,8] (Supplementary Fig. 1). The multi-year persistence of these El Niño and La Niña events exacerbates their induced climate impacts, causing persistent floods and droughts worldwide[9–11].

The occurrence of multi-year ENSO events results in non-cyclic evolution of ENSO and poses a challenge to the prediction of ENSO. For example, the 2014/15/16 multi-year El Niño event surprised the ENSO scientific community by its unique evolution, and most operational forecasting models compiled at the International Research Institute for Climate and Society (IRI) failed to predict the evolution of this event[12] (Supplementary Fig. 2). To successfully forecast multi-year ENSO events requires an understanding of the underlying physical mechanisms that drive their unique evolutions. However, despite the many hypotheses that have been proposed to explain the asymmetric duration of El Niño and La Niña (that is, La Niña tends to be more persistent than El Niño)[13–18], few theories exist to account for the sources of the multi-year persistence of El Niño and La Niña[7,19–23]. Moreover, almost all existing theories emphasize the importance of ocean–atmosphere coupled processes within the tropical Pacific, Indian, and Atlantic Oceans for the generation of multi-year ENSO events. In contrast, extratropical forcings are not considered as a key source of multi-year ENSO events. However, recent studies have suggested that extratropical atmospheric variability, such as the North Pacific Oscillation (NPO)[24,25], may play an important role in affecting the development[26–29], pattern[30–32], phase transition[33–35], and decadal variability[36] of ENSO. Despite these extensive studies, it remains unclear if the NPO and multi-year ENSO events are dynamically linked, and if they are, then how.

Here we use observational analyses in combination with a series of climate model experiments to show that persistent two-way teleconnections between the NPO and the tropical Pacific constitute a key source of multi-year El Niño events. The NPO during the boreal winter can induce a Central Pacific (CP) El Niño[37–40] during the subsequent winter[30], which in turn feeds back into the North Pacific to re-intensify the variability in the NPO, thereby maintaining El Niño conditions for another year. Our results highlight the importance of extratropical atmospheric variability in generating multi-year El Niño events.

## Results

### Observed linkage between the NPO and multi-year El Niño.
The five multi-year El Niño events observed since 1950 (Supplementary Table 1; see Methods for the definition of multi-year El Niño events) are used to examine the role of NPO atmospheric forcing in driving multi-year El Niño events. All the sea level pressure (SLP) anomalies during the previous winter (January–February–March, JFM) before these multi-year El Niño events show a north–south dipole feature in the North Pacific (Supplementary Fig. 3a–e), which is typical of the NPO pattern[26,27]. We denote the year when El Niño first develops as year (0), and the following two years as years (1) and (2), respectively. It is noted that the JFM(0) NPO index is greater than 1.0 in all five multi-year El Niño events (Fig. 1a and

Supplementary Fig. 3g), and the spatial correlation coefficients of the JFM(0) SLP patterns before the five multi-year El Niño events with the typical NPO pattern (Supplementary Fig. 3f) are all significant above the 95% confidence level (Supplementary Fig. 3h). Taking the latest 2018/19/20 El Niño event as an example, a distinct NPO-like SLP anomaly pattern was present during the boreal winter (JFM) of 2018 in the North Pacific, having a strong positive NPO index of 1.27 and a high spatial correlation coefficient of 0.81. This suggests that the NPO may have played a key role in the occurrence of multi-year El Niño events.

### Two-way feedback mechanism between the tropics and extratropics.
Previous studies have shown that the North Pacific meridional mode (NPMM)[41] is an effective conduit through which the NPO can eventually affect the tropics and ENSO variability[30,42–44]. It has been demonstrated that the NPO can reduce surface evaporation and increase SST in the subtropical northeastern Pacific by reducing the speed of the subtropical northeasterly trade winds. The warming of the subtropical northeastern Pacific would further reduce the trade winds and initiate a positive thermodynamic feedback among surface winds, evaporation, and SST, known as the wind–evaporation–SST (WES) feedback[45]. This WES feedback excites the NPMM, which propagates positive SST anomalies from the subtropics into the central equatorial Pacific, where positive SST anomalies are conducive to the development of El Niño. In addition to the NPMM, there are at least two other mechanisms by which NPO atmospheric anomalies can lead to the onset of El Niño through the weakening of off-equatorial trade winds[46] or the excitation of the off-equatorial Rossby wave[47]. However, studies of these ENSO extratropical precursor dynamics have focused mainly on the effects of the NPO on the onset of El Niño rather than on the potential role of the NPO in prolonging the duration of El Niño. Thus, there is a need to develop an understanding of the dynamics underlying the link between the NPO and multi-year El Niño events.

Figure 2 shows the evolutions of SST and SLP anomalies composited for five multi-year El Niño events over a 3-year period. The positive NPO forcing during JFM(0) (Fig. 2b) generates positive SST anomalies extending from the subtropical northeastern Pacific to the central equatorial Pacific, resembling a positive phase of the NPMM[43] (Fig. 2a), which maintains positive SST anomalies for several seasons and extends them equatorward into the central equatorial Pacific, finally leading to equatorial Pacific warming during the subsequent JFM(1) (Fig. 2c; see Supplementary Fig. 4 for more details). This process involves teleconnections from the extratropics to the tropics (termed "extratropical−tropical teleconnections"), consistent with previous findings[26–30].

We then examine whether the equatorial Pacific warming can exert an influence on the extratropics. We note that the NPO-induced Pacific SST anomaly pattern during JFM(1) resembles that of the CP El Niño[37–40], characterized by the center of the warming being located mainly in the central equatorial Pacific (Fig. 2c; see also Supplementary Fig. 5a). It has been recognized that the extratropical SLP response to ENSO is sensitive to the longitudinal location of maximum SST anomalies along the equatorial Pacific[48]. Unlike the projection of the Eastern Pacific (EP) El Niño onto the Aleutian Low, the CP El Niño projects onto a different SLP pattern in the North Pacific, and is associated with negative SLP anomalies over the Hawaiian region[49,50] (Supplementary Fig. 6). The Hawaiian SLP variability, represented by the time series of SLP anomalies averaged over the Hawaiian region (SLP$_{HI}$; 155°–125°W, 16°–32°N; red box in Fig. 2d), is linked to the NPO-like SLP variability and

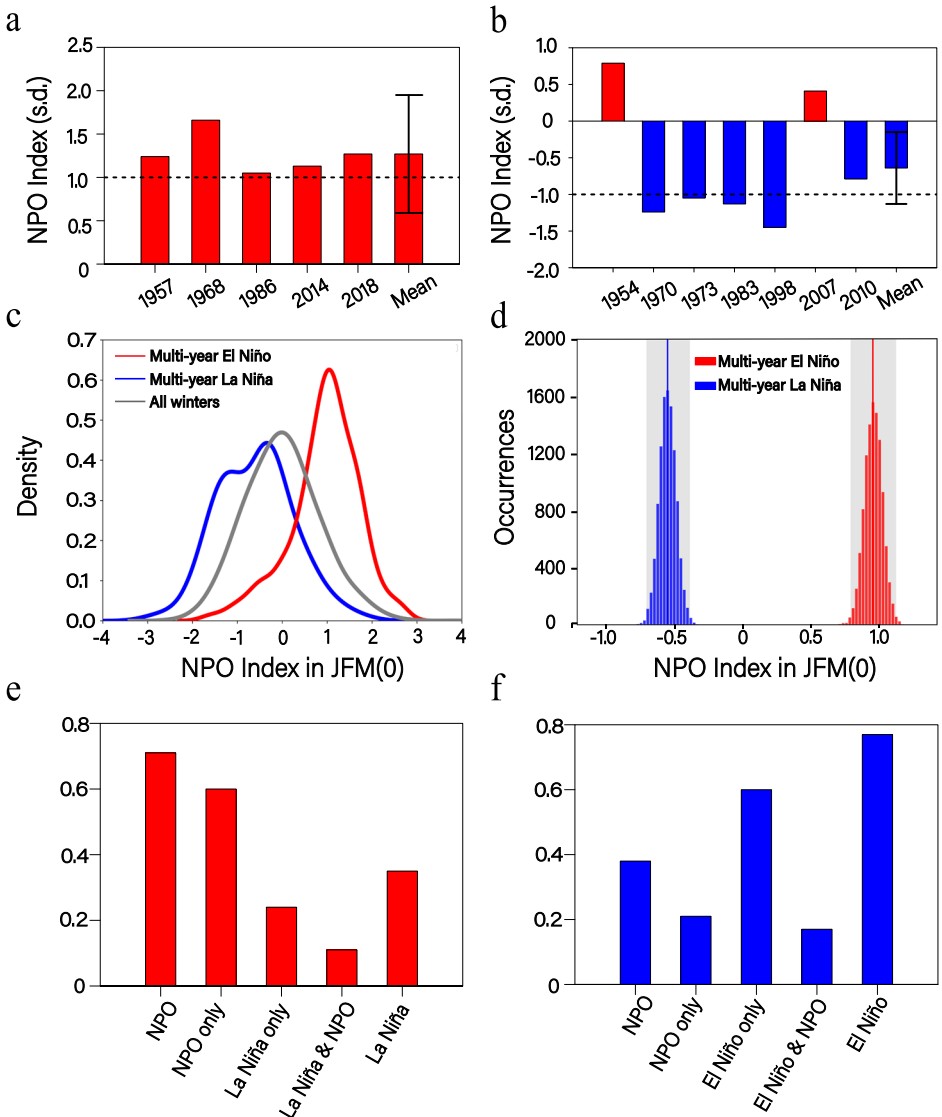

**Fig. 1 The NPO states prior to multi-year El Niño and La Niña events in the observations and models.** Normalized JFM(0) NPO indices for **a** the five selected multi-year El Niño events and **b** the seven selected multi-year La Niña events in observations. Error bars in the ensemble mean indicate the 95% confidence interval, and horizontal dashed lines represent one positive and one negative standard deviation in **a**, **b**, respectively. **c** Probability density functions (PDFs) of the normalized JFM(0) NPO indices for multi-year El Niño events (red curve), multi-year La Niña events (blue curve), and all winters (gray curve) in the CMIP5/6 models. **d** Histograms of 10,000 realizations of the bootstrap method for the normalized JFM(0) NPO indices of multi-year El Niño (red) and La Niña (blue) events in the CMIP5/6 models. Vertical red and blue lines indicate the mean values of 10,000 inter-realizations for the boreal winter NPO index before multi-year El Niño events and multi-year La Niña events, respectively. Gray shaded regions indicate the respective doubled standard deviations (SDs; the 95% confidence interval based on the normal distribution) of the 10,000 inter-realizations (see Methods). **e** Ratios of multi-year El Niño events preceded by NPO events (NPO), NPO without La Niña events (NPO only), La Niña occurring without NPO events (La Niña only), La Niña with NPO events (La Niña & NPO), and La Niña events (La Niña) during JFM(0) in the CMIP5/6 models. **f** As in **e**, but for the ratios of multi-year La Niña events preceded by NPO, NPO only, El Niño only, El Niño & NPO, and El Niño, respectively.

NPMM (Supplementary Fig. 7). The composite $SLP_{HI}$ and NPO indices for multi-year El Niño events show two peaks: one during JFM(0) and the other during JFM(1) (Fig. 3a, b). The re-intensification of the $SLP_{HI}$ and NPO indices during JFM(1) is consistent with the warming in the central tropical Pacific associated with CP El Niño, implying that the NPO-induced central tropical Pacific SST variability can feedback into the North Pacific, generating negative SLP anomalies over the Hawaiian region (by extension of the positive phase of the NPO in the North Pacific). This process is referred to as tropical−extratropical teleconnections.

The anomalous low pressure over the Hawaiian region in response to central equatorial Pacific warming during JFM(1)

produces an anomalous southwesterly flow on its eastern flank (Supplementary Fig. 4e), which acts to re-weaken the off-equatorial trade winds, and in turn activate ENSO precursor dynamics such as the NPMM[41], resulting in a re-intensification of the NPMM during March–April–May (MAM)(1) (Fig. 3c). This positive feedback between the NPMM and CP El Niño on interannual timescales is consistent with the recent findings of Stuecker[51] and Fang and Yu[52]. Through NPMM dynamics, positive SST anomalies in the subtropical northeastern Pacific persist through June–July–August (JJA)(1) and propagate gradu-ally from the subtropics into the central equatorial Pacific[30,42–44] (Supplementary Fig. 4), thereby re-initiating the development of El Niño of year (1) (Fig. 2e) and resulting in multi-year El Niño

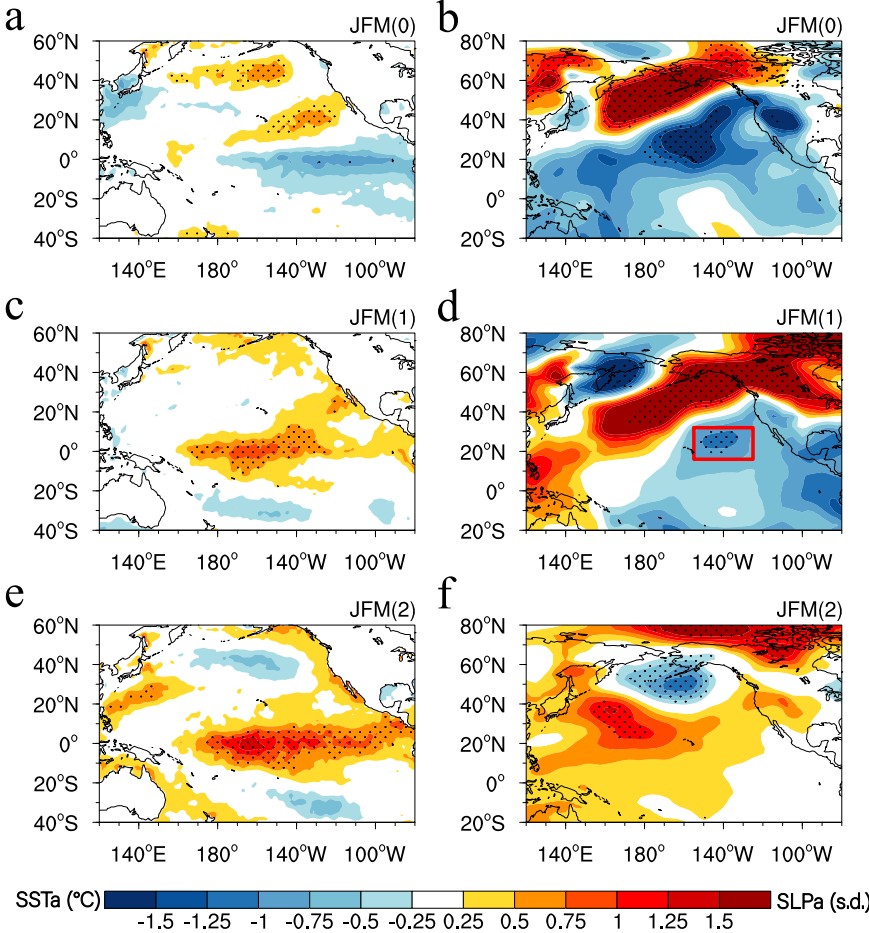

**Fig. 2 Evolutions of SST and SLP anomalies composited for multi-year El Niño events over a 3 yr period derived from HadISST and NCEP1 SLP datasets. a, b** JFM(0) SST and SLP anomalies, respectively. **c, d** JFM(1) SST and SLP anomalies, respectively. **e, f** JFM(2) SST and SLP anomalies, respectively. The red box in **d** denotes the region used to compute the SLP index over the Hawaiian region (SLP$_{HI}$; 155°–125°W, 16°–32°N). In **a−f**, dots indicate SST and SLP anomalies significant at the 95% confidence level.

events. The results are robust among different SST and SLP data sets (Supplementary Fig. 8).

Through a careful examination of each of the individual multi-year El Niño events (Supplementary Fig. 9), we note that the individual evolution of each event is similar to the composite results presented above. That is, the positive NPO event (defined by a 1.0 SD threshold of the JFM(0) NPO index) is followed by a typical CP El Niño pattern[37–40] or a mixed pattern[40] of CP and EP El Niño in JFM(1), characterized by maximum warming near the central equatorial Pacific. Furthermore, there are negative SLP anomalies between Hawaii and western North America that are consistent with a typical pattern of the SLP response to CP El Niño SST forcing. The negative SLP anomalies weaken the off-equatorial trade winds and then strengthen the NPMM, resulting in the multi-year evolution of El Niño.

The schematic of Fig. 4 presents a two-way feedback mechanism between the tropics and extratropics associated with the CP El Niño phenomenon[37–40] to explain the dynamics underlying the linkage between the NPO and multi-year El Niño events. The boreal winter NPO induces a CP-type El Niño event over the equatorial Pacific during the subsequent winter through its effect on the NPMM[41]. This CP-type El Niño in turn feeds back into the North Pacific to force changes in atmospheric circulation over the Hawaiian region, which re-activate the NPMM to favor the development of another El Niño event[52]. This process continues until it is disrupted by negative feedbacks,

as suggested by the ENSO cycle[53] or noise in the air–sea coupled system[54]. The two-way feedback mechanism between the tropics and extratropics proposed here to explain the sources of multi-year El Niño events echoes that previously identified by Di Lorenzo et al.[31] to explain the sources of tropical Pacific decadal (timescale >8 years) variability. Di Lorenzo et al.[31] suggested that the NPO atmospheric variability, which acts as the extratropical stochastic forcing, may enhance the low-frequency variance of ENSO through a chain of extratropical−tropical feedback processes. The present study demonstrates that a similar dynamical chain may also validly explain the dynamics underlying the emergence of multi-year El Niño events, which are dominated primarily by interannual variations. Taken together, it would appear that NPO atmospheric forcing, through extratropical−tropical feedbacks, may contribute to enhancing the low-frequency variability in ENSO over a wide spectrum of time scales (interannual-to-decadal timescales).

The schematic of Fig. 4 highlights the potential importance of CP El Niño in linking the NPO to multi-year El Niño events. Previous studies have suggested that the NPO tends to induce the CP-type El Niño, but not in all cases[29]. Equatorial ocean dynamics, such as zonal advection in the tropical Pacific, can extend NPO-induced warming in the central equatorial Pacific eastwards, leading to EP El Niño events[55]. Several single-year El Niño events (1965/66, 1972/73, 1991/92, and 1997/98), which are also preceded by a positive NPO

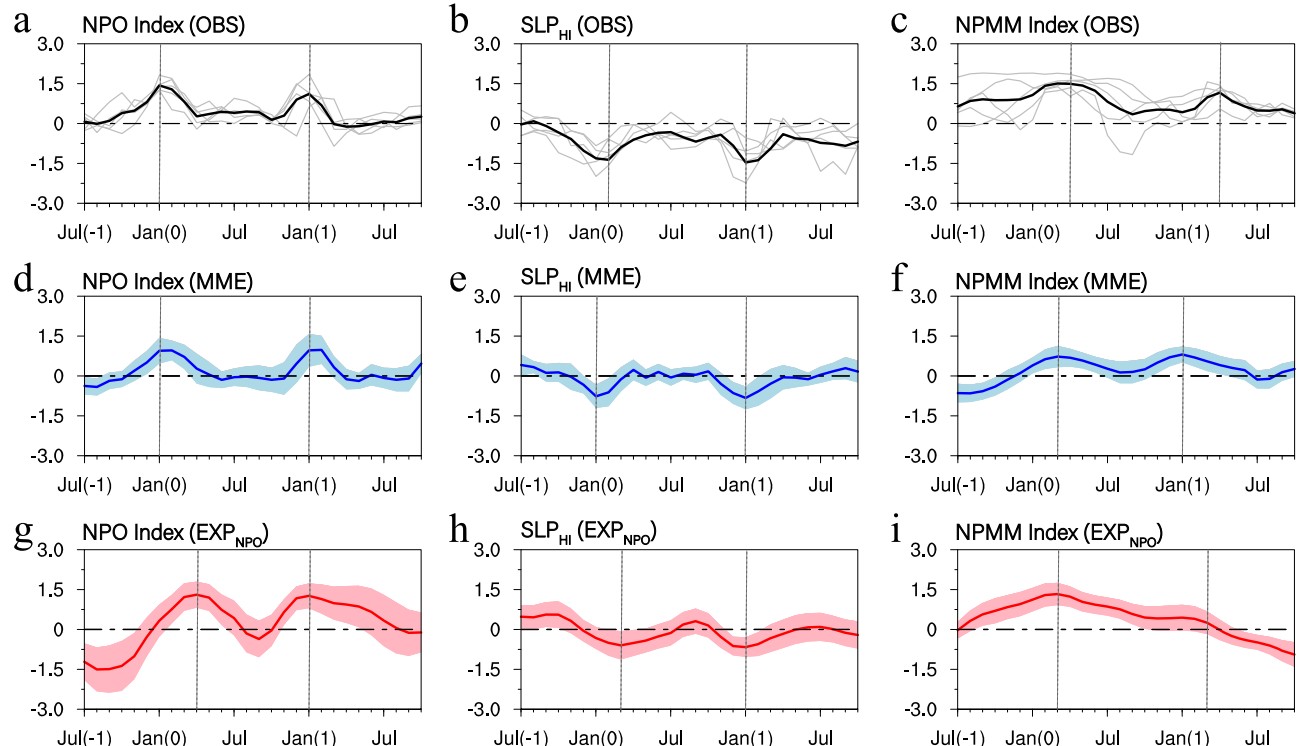

**Fig. 3 Temporal evolutions of various indices over the North Pacific for multi-year El Niño events. a–c** Temporal evolutions of the NPO index, the SLP index over the Hawaiian region (SLP$_{HI}$), and the NPMM index from July(−1) to October(1) for the observed multi-year El Niño events, respectively. **d–f** As in **a–c**, but for multi-year El Niño events in the CMIP5/6 models. **g–i** Ensemble-mean difference in the NPO, SLP$_{HI}$, and NPMM indices from July(−1) to October(1) between the positive NPO forcing and CTRL experiments. In **a–c**, the gray curves indicate individual evolutions of the observed multi-year El Niño events, and the black curve indicates the mean of the gray curves. In **d–i**, the colored shading indicates interquartile ranges between the 25th and 75th percentiles. In **a–i**, the vertical dashed lines denote the time when various indices reach the peak.

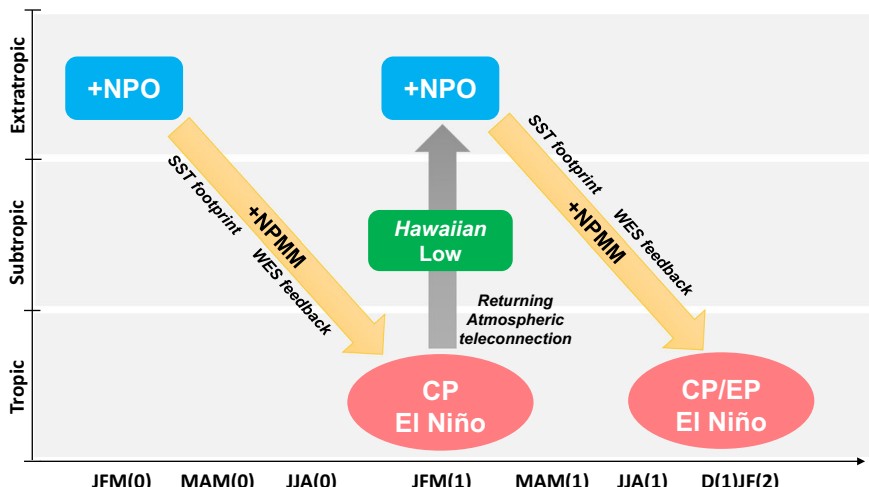

**Fig. 4 Schematic diagram illustrating the two-way feedback mechanism between the tropics and extratropics.** The positive NPO forcing during JFM(0) can induce an SST footprint in the subtropical northeastern Pacific during MAM(0) by changing the northeasterly trade winds, which resembles a positive phase of the NPMM. The NPMM interacts with the trade winds and extends positive SST anomalies equatorward into the central equatorial Pacific through the WES feedback, leading to a CP-type El Niño event over the equatorial Pacific during the subsequent JFM(1). This CP-type El Niño in turn feeds back into the North Pacific to force changes in the atmospheric circulation over the Hawaiian region that project onto the NPO variability, which re-activates the NPMM to favor the development of another CP-type or EP-type El Niño event. Through this two-way feedback process between the tropics and extratropics, the NPO contributes to multi-year persistence of El Niño events in the tropical Pacific.

event during JFM(0) (Supplementary Fig. 10d), are characterized by a typical EP El Niño pattern during JFM(1) (Supplementary Fig. 10b). Consistent with the location of maximum warming, there is only a weak positive SLP response over the Hawaiian region (Supplementary Fig. 10e, h). As a result, the NPMM is very weak (|NPMM index| < 0.5 SD) during MAM(1) (Supplementary Fig. 10i), and the warming in the equatorial central−eastern Pacific evolves rapidly to a La Niña-like condition by JFM(2) (Supplementary Fig. 10c), thereby leading to a single-year El Niño event.

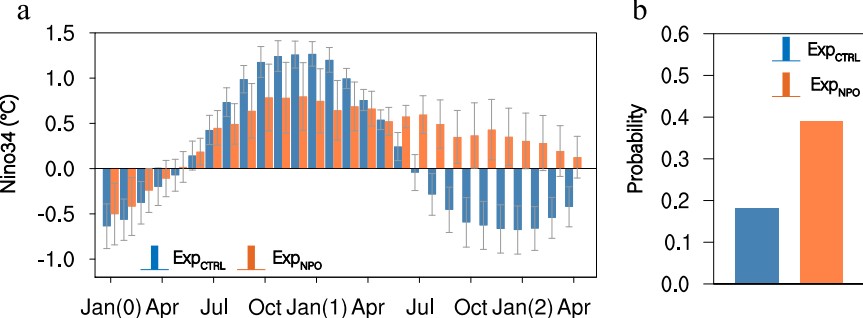

**Fig. 5 The impact of the NPO on the evolution of El Niño events in the CGCM experiments. a** Ensemble-mean differences in the Niño3.4 index from January(0) to April(2) between the positive NPO forcing and CTRL experiments (red bars). As a contrast, also shown is the ensemble-mean Niño3.4 index of El Niño events in the CTRL experiment (green bars). Error bars indicate the 95% confidence intervals. **b** Probability of multi-year El Niño events in the positive NPO forcing (red bar) and CTRL (green bar) experiments based on 60 ensemble member simulations.

Although CP El Niño is important in bridging the NPO to multi-year El Niño events, this does not mean that all CP El Niño events can evolve as a multi-year event. For the NPO-preceded CP El Niño events, positive SST anomalies extend more westward (west of 170°E) into the western Pacific warm pool where the deep convection can more easily be excited[35] (Supplementary Fig. 11a), thereby favoring a stronger SLP response over the Hawaiian region (Fig. 3b). In contrast, for the non-NPO-preceded CP El Niño events (1977/78, 1994/95, 2002/03, and 2009/10), positive SST anomalies are confined mainly east of 170°E and do not extend sufficiently into the western Pacific warm pool (Supplementary Fig. 11b). Therefore, the non-NPO-preceded CP El Niño events are all accompanied by a weaker SLP response over the Hawaiian region during JFM(1) (Supplementary Fig. 12h) and a weaker NPMM-related SST anomaly band during MAM(1) (Supplementary Fig. 12i). As a result, El Niño conditions are not maintained for an additional year (Supplementary Fig. 12c), leading to single-year evolution. The results presented above suggest that both the NPO and its induced CP El Niño are necessary for generating multi-year El Niño events, consistent with the dynamical hypothesis that we have proposed (Fig. 4).

**Simulated impacts of the NPO on multi-year El Niño events.** The role of the NPO in generating multi-year El Niño events is also supported by historical (1900–1999) simulations of CMIP5/6 models (Supplementary Table 3; see "Methods" for model selections). Of the 29 selected models, 26 generate a high occurrence ratio (>60%) of multi-year El Niño events that are preceded by positive NPO events (Supplementary Fig. 13a). The multi-model ensemble (MME) result indicates that 71% of multi-year El Niño events are preceded by positive NPO events, whereas only 29% of multi-year El Niño events are not linked to the preceding positive NPO events. The probability of the NPO-preceded multi-year El Niño events (71%) is significantly higher than the probability of the NPO-preceded El Niño events (including both single-year and multi-year El Niño) (41%) and the probability of the NPO-preceded single-year El Niño events (27%) (Supplementary 14), indicating that in contrast with single-year El Niño, multi-year El Niño is more closely tied to the precedent NPO. For all selected CMIP5/6 models, a pronounced shift in the probability distribution toward positive values of the JFM(0) NPO is obvious for multi-year El Niño (Fig. 1c). The composited value (+0.95) of the JFM(0) NPO index for multi-year El Niño is statistically significant above the 95% confidence level, according to the bootstrap method (see Methods and Fig. 1d). These CMIP5/6 model results provide additional evidence that extratropical atmospheric variability associated with the NPO contributes significantly to the generation of multi-year El Niño events.

The composited evolutions of the NPO-preceded multi-year El Niño events in the CMIP5/6 models (Supplementary Figs. 5b and 15) also support the mechanism from observations, namely, that NPO-induced central equatorial Pacific warming during JFM(1) excites atmospheric teleconnections to the extratropics that re-energize the NPO-like SLP variability and then re-activate the NPMM (Fig. 3d–f), which maintains the equatorial Pacific warming until JFM(2) and produces multi-year El Niño events.

To further verify the role of the NPO forcing, two model experiments were conducted using a coupled general circulation model (CGCM) with and without the positive NPO forcing (EXP$_{NPO}$ and EXP$_{CTRL}$, respectively). The imposed NPO forcing is represented by the observed air temperature (T) and wind (U, V) anomalies associated with the NPO, which are assimilated to constrain the atmospheric component in the CGCM model (see Methods and Supplementary Fig. 16). The model results confirm that the positive NPO forcing tends to prolong the duration of El Niño events during the subsequent winter (Fig. 5a). El Niño events forced by the NPO are characterized by a pattern of SST anomalies that resemble those of the CP El Niño[37–40], with the maximum warming near the central Pacific (Supplementary Fig. 5c). The warming in the central equatorial Pacific in turn forces changes in SLP anomalies in the central North Pacific near Hawaii (Fig. 3h) that re-energize the NPO-like SLP variability (Fig. 3g), which then enhances the NPMM (Fig. 3i) to delay the termination of El Niño events and to prolong their duration. Owing to the presence of the NPO forcing, multi-year El Niño events become more frequent in the EXP$_{NPO}$. Specifically, the occurrence ratio of multi-year El Niño events is increased from 18% in the EXP$_{CTRL}$ to 39% in the EXP$_{NPO}$ (Fig. 5b). This increase is statistically significant above the 95% confidence level, according to the bootstrap method. These modelling results support the hypothesis that NPO variability could contribute to the genesis of multi-year El Niño events.

**Projections of future climate change.** Although multi-year El Niño events have occurred only five times since 1950, they have occurred twice during the last decade (Supplementary Fig. 1). This raises the question as to whether multi-year El Niño events might become more frequent under anthropogenic forcing-induced warming, particularly in the twenty-first century and beyond. To address this question, we compared the number of multi-year El Niño events that occurred in the present (1900–1999) and future (2000–2099) climates using the 23 selected CMIP5/6 models (see "Methods" and Supplementary Table 3). The present and future climates are represented by the historical and representative concentration pathway 8.5 (RCP8.5) experiments, respectively. 16 of the 23 models (70%) simulate an increased frequency of multi-year El Niño events in the future

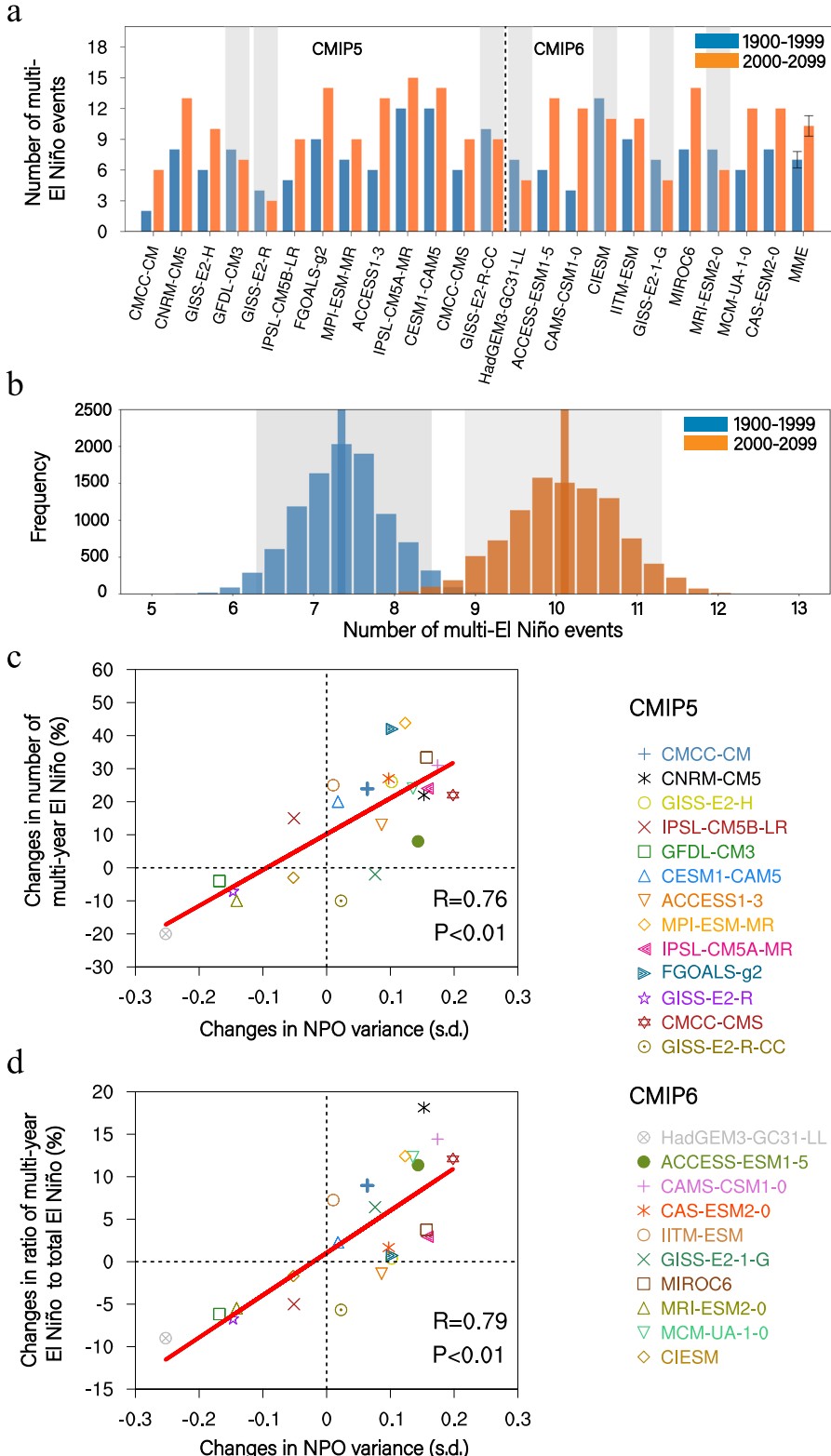

warmer climate, with a good inter-model consensus (Fig. 6a). The ensemble mean increase is 43% (from about 7 events per 100 years in the present climate to 10 events per 100 years in the future climate), which is significant above the 95% confidence level according to a bootstrap test (Fig. 6b).

Previous studies suggested that the NPO variance may increase in a warmer climate either due to a more energetic WES feedback[56] or due to a stronger NPO excitation driven by the Kuroshio Extension downstream atmospheric response[57,58]. Given the significant impact of the NPO on multi-year El Niño events, we hypothesized that more frequent occurrences of multi-year El Niño events may be linked to increased NPO variance in the future climate. As initial support for this hypothesis, an inter-model relationship shows that models with the increased variance

**Fig. 6 Relationship between future change in multi-year El Niño event frequency and NPO variance. a** The number of multi-year El Niño events that occurred in the present (1900–1999, blue) and future (2000–2099, red) climates. The seven models that simulate a decrease in number are grayed out. Error bars in the multi-model mean correspond to the 95% confidence interval. **b** Histograms of 10,000 realizations of a bootstrap method for the number of multi-year El Niño events in the present (blue) and future (red) climates. The blue and red lines indicate the mean values of the 10,000 realizations for the present and future climates, respectively. The gray shaded areas refer to the respective 1.0 SD of the 10,000 realizations. **c** Inter-model relationship between the change in the number of multi-year El Niño events and the change in the NPO variance. The linear regression line is shown by the solid line, with the significant correlation coefficient (R) and significance level (P) being indicated in the right bottom corner of the figure. **d** As in **c**, but for the inter-model relationship between the change in the ratio of the number of multi-year El Niño events to the total number of El Niño events and the change in the NPO variance. The change in the number of multi-year El Niño events is defined as $\Delta = (m_2 - m_1)/m_1$, where $m_1$ and $m_2$ represent the numbers of multi-year El Niño events that occurred in the present and future climates, respectively. The change in the ratio of the number of multi-year El Niño events to the total number of El Niño events is defined as $\Delta = m_2/n_2 - m_1/n_1$, where $n_1$ and $n_2$ represent the total number of El Niño events that occurred in the present and future climates, respectively.

of the NPO index in the future climate systematically produce a more frequent occurrence of multi-year El Niño events, and this tendency is statistically significant (Fig. 6c). Furthermore, the ratio of the number of multi-year El Niño events to the total number of El Niño events is also projected to increase due to increased NPO variance (Fig. 6d), regardless of how the total number of El Niño events changes in response to anthropogenic global warming. Although further studies are required to determine how the NPO dynamics are responding to a warmer climate, these results suggest that an increase in NPO variance might not only energize the ENSO system[56] but also might lead to more frequent multi-year El Niño events.

## Discussion

In summary, we have shown that the NPO atmospheric forcing, which tends to prolong the duration of El Niño events, is a key source of multi-year El Niño events. The distinctive role of the NPO is its ability to trigger the NPMM and subsequently CP El Niño during the subsequent winter. Once triggered by the NPO, CP El Niño can excite atmospheric teleconnections to the North Pacific that re-energize the NPO-like SLP variability, which then re-activates the NPMM to favor the development of another El Niño event, leading to multi-year El Niño events. In association with the increased NPO variability, multi-year El Niño events may occur more frequently in the future warmer climate. Our study differs from previous studies[22,23] by attributing one of the dominant sources of multi-year El Niño to NPO-like extratropical atmospheric variability rather than to tropical ocean–atmosphere coupled variability. Our findings draw attention to the extratropical influence on the duration of El Niño and have important implications for the prediction and projection of multi-year El Niño events.

While the North Pacific precursor patterns for multi-year El Niño events have a strong NPO/NPMM structure, we note that in observations there is also an evident La Niña-like condition in the precursor patterns (Fig. 2a). Given the few realizations in the observational record, the dynamical significance of the tropical Pacific cooling is still unclear. In the CMIP5/6 models, the probability of NPO events co-occurring with La Niña prior to multi-year El Niño events is relatively low (~10%; Fig. 1e). The multi-model ensemble precursor patterns also exhibit a weak cooling in the tropical Pacific, but its amplitude is much smaller compared to the strength of the NPO signal (Supplementary Fig. 15a). In this regard, further modelling studies are required to explore the relative importance of the NPO and La Niña states in developing subsequent multi-year El Niño events.

Our result reveals a strong influence of the NPO on multi-year El Niño events. A question arises as to whether the NPO also has a substantial influence on multi-year La Niña events. We find that there is a relatively high likelihood of multi-year La Niña events in observations (57%; Supplementary Fig. 17) and CMIP5/6

models (77%; Fig. 1f) that are preceded by strong El Niño events in the tropical Pacific, consistent with previous findings that the duration of La Niña events is strongly affected by the strength of preceding El Niño events through the recharge process of the equatorial oceanic heat content[7,59]. In contrast, only 21% of multi-year La Niña events are preceded by negative NPO events alone without an accompanying strong El Niño in models (Fig. 1f). Thus, multi-year La Niña events may differ from multi-year El Niño events with respect to the connection with the precedent NPO (Fig. 1e, f).

Nevertheless, the composite analysis shows that a negative NPO event alone during JFM(0) is followed by a distinct La Niña pattern in JFM(1) whose anomalous cold SST extends to the central equatorial Pacific region (Supplementary Fig. 18a), which establishes atmospheric teleconnections to the North Pacific that lead to the re-emergence of positive SLP anomalies over the Hawaiian region (Supplementary Fig. 18c). Owing to the re-emergence of these positive SLP anomalies, a negative NPMM characterized by negative SST anomalies over the subtropical northeastern Pacific begins to re-intensify around JFM(1) (Supplementary Fig. 18d), ultimately resulting in multi-year La Niña events. These results suggest that although the phase transitions of La Niña are determined largely by the recharge process of equatorial oceanic heat content, as ENSO theories suggest[60], the NPO may also play a role in developing multi-year La Niña events, resembling its role in developing multi-year El Niño events. It is likely that the preceding NPO and El Niño may, together or separately, influence the occurrence of multi-year La Niña events (Fig. 1b, f and Supplementary Fig. 17).

In this study, we emphasize the role of the NPO in generating multi-year ENSO events but do not exclude the contribution of other processes to the duration of ENSO. Many other internal and external factors driving ENSO dynamics—from the tropical Pacific[53], tropical Atlantic[61], tropical Indian[34], or South Pacific Oceans[62–64]—may also affect the duration of ENSO. For example, some studies have suggested that forcing from the southeastern subtropical Pacific could also have an impact on the evolutions of the 2014/15/16 El Niño event[65,66]. The feedbacks among these processes, as well as their relative importance, need to be further clarified to achieve a comprehensive understanding of the primary factors controlling the duration of ENSO events.

## Methods

**Observation-based data.** The analyzed monthly SST data are from: (1) the Hadley Center Sea Ice and SST dataset version 3 (HadISST)[67]; (2) National Oceanic and Atmospheric Administration Extended Reconstructed SST version 4 (ERSST)[68]; (3) Kaplan Extended SST version 2 (Kaplan SST)[69]; and (4) Centennial in situ Observation-Based Estimates SST version 2.9.2 (COBE SST)[70]. The monthly atmospheric fields including surface winds and SLP were from the National Center for Environmental Prediction (NCEP)/National Center for Atmospheric Research (NCAR) reanalysis 1 (NCEP1)[71]. The SLP field derived from the Hadley SLP data set (HadSLP2)[72] was also employed to test the reliability of the results. All of the

anomalies in this study are relative to the climatological annual cycle for the period 1981–2010.

**CMIP5/6 simulations under historical and future scenarios**. To examine the contributions of the NPO to multi-year El Niño/La Niña events, we analyzed the historical (1900–1999) simulations from coupled general circulation models (CGCMs) participating in the CMIP5[73] and CMIP6[74]. According to our hypothesis, the influences of the NPO on multi-year El Niño and La Niña events involve a chain of tropical−extratropical feedback processes associated with the CP El Niño phenomenon. To successfully simulate the role of the NPO in developing multi-year El Niño and La Niña events, the model should be able to simulate both the NPO–ENSO connection and the two types of El Niño. Therefore, we first correlated the JFM(0) NPO index with the following JMF(1) Niño3.4 index over the 100-year period using the CMIP5/6 historical simulations. A total of 51 CMIP5/6 models simulate a significant connection reminiscent of the observations. We then used the correlation coefficient between the Niño3 index (150°W to 90°W, 5°S to 5°N) and the Niño4 index (160°E to 150°W, 5°S to 5°N) during the DJF season of El Niño events in the pre-selected 51 models to evaluate whether these models can simulate the two types of El Niño[75]. The correlation between the Niño3 and Niño4 indices in the CMIP5/6 models varies from −0.46 to 0.88, compared with an observed correlation of 0.48 for the period 1900–1999. Finally, we selected the 29 models (Supplementary Table 3) whose correlation coefficients were close to that of the observations. We used these 29 selected CMIP5/6 historical simulations to examine the role of the NPO in generating multi-year El Niño/La Niña events.

Simulations under the historical (1900–1999) and representative concentration pathway (RCP) 8.5 (2000–2099) scenarios from CMIP5/6 were also used to examine the relationship between future change in the frequency of multi-year El Niño events and change in the NPO variability. Of the selected 29 models based on their historical simulations, there are 23 CMIP5/6 models whose RCP8.5 simulations are available (Supplementary Table 3). Therefore, we used these 23 models to analyze the relationship under the RCP8.5 scenario.

**Definition of multi-year El Niño/La Niña events**. For the observational analysis, multi-year El Niño and La Niña events are defined based on monthly SST anomalies averaged over the Niño3.4 region (5°S to 5°N, 120° to 170°W; termed the "Niño3.4 index") for 1950–2020. The monthly Niño3.4 index is linearly detrended and smoothed with a 3-month-running-mean filter. We denote the year when El Niño and La Niña first develop as year (0), and the following two years as year (1) and year (2), respectively. A multi-year El Niño (La Niña) event is identified when the Niño3.4 index is over 0.75 (below −0.75) standard deviations in any month during October(0) to February(1), and remains above zero thereafter and exceeds 0.5 (−0.5) standard deviations again in any month during October(1) to February(2)[7]. According to these criteria, there are five multi-year El Niño events and seven multi-year La Niña events for 1950−2020 (Supplementary Table 1). The remaining El Niño (La Niña) events, which transitioned to a La Niña-like (El Niño-like) condition in year (1), are considered as transitional single-year events (Supplementary Table 2). The definitions of multi-year El Niño and La Niña events are the same for both models and observations.

**NPO, NPMM, and CP ENSO indices**. The NPO mode is generally defined as the second leading empirical orthogonal function (EOF2) of boreal winter (JFM) SLP anomalies in the North Pacific poleward of 15°N[24,25]. The NPO index is defined as the second principal component (PC2) time series associated with the EOF2. The NPMM represents a coupled SST-surface wind pattern over the subtropical northeastern Pacific[41]. The NPMM index is defined as the normalized SST anomalies over the subtropical northeastern Pacific (15°–25°N and 150°–120°W)[76]. The CP ENSO pattern and index are defined by computing the first empirical orthogonal function (EOF1) and principal component (PC1) of tropical Pacific SST anomalies between 20°S and 20°N after subtracting the anomalies regressed on the Niño1 + 2 index (representing the influence of the canonical ENSO)[39], respectively. Similarly, the EP ENSO pattern and index are defined by computing the EOF1 and PC1 of tropical Pacific SST anomalies between 20°S and 20°N after subtracting the anomalies regressed on the Niño4 index (representing the influence of the CP ENSO), respectively.

**Statistical significance test**. We used a bootstrap method[77] to examine whether the boreal winter NPO index before multi-year El Niño events and multi-year La Niña events is significantly different. The 135 values of the JFM(0) NPO index for multi-year El Niño events and 182 values of the JFM(0) NPO index for multi-year La Niña events in the historical simulations from the 29 selected CMIP5/6 models were re-sampled randomly to construct 10,000 realizations of the NPO index over the 29 models, respectively. Any model can be selected again in this random resampling process. The red and blue vertical lines in Fig. 1d indicate the mean values of 10,000 inter-realizations for the boreal winter NPO index before multi-year El Niño events and multi-year La Niña events, respectively. The gray shaded regions in Fig. 1d indicate the respective doubled standard deviations (SDs) (the 95% confidence interval based on the normal distribution) of the 10,000 inter-realizations. If the gray shaded regions do not overlap each other, then the statistical significance is above the 95% confidence level. The statistical significance of

correlations/regressions was calculated using a two-tailed Student's $t$-test with $n − 2$ degrees of freedom ($n$ being the number of years).

**Model description**. The model used in sensitivity experiments of this study is the Flexible Global Ocean–Atmosphere–Land System Model Grid-point version 2 (FGOALS-g2), one of the CMIP5 models, developed by the State Key Laboratory of Numerical Modeling for Atmospheric Sciences and Geophysical Fluid Dynamics (LASG), Institute of Atmospheric Physics (IAP), Chinese Academy of Sciences (CAS)[78]. The FGOALS-g2 is a state-of-the-art CGCM, consisting of the Grid-point Atmospheric Model of LASG/IAP version 2 (GAMIL2), the LASG/IAP Climate System Ocean Model version 2 (LICOM2), the Community Land Model version 3 (CLM3), and the improved version of Community Ice CodE version 4 by LASG (CICE4-LASG) connected using the coupler (CPL6). The atmospheric component has a resolution of about 2.8° × 2.8° in the horizontal dimension and 26 layers in the vertical dimension up to 2.194 hPa. The oceanic component has a horizontal resolution of 1° × 1° (with 0.5° in the meridional direction in tropical areas) and has 30 vertical levels with an interval of 10 m in the upper 145 m and increasing with depth.

Recently, a weakly coupled data assimilation (WCDA) system for constraining the atmospheric component in a coupled model was developed and applied to the FGOALS-g2 model[79]. The data assimilation method used in the WCDA system is the Dimension Reduced Projection Four-Dimensional Variational (DRP-4DVar) scheme[80], which is an economical approach for implementing 4DVar by using the technique of dimension reduced projection (DRP). Although the ocean component of the coupled model is not assimilated, the WCDA system can transfer the assimilated atmospheric information to the ocean through air–sea coupling. Thus, we tested the role of the North Pacific NPO forcing by conducting experiments in which a positive NPO forcing is only added to the atmosphere component of the FGOALS-g2 model.

**Sensitivity experiments**. The imposed NPO forcing is represented by monthly air temperature (T) and wind (U, V) anomalies at three low pressure levels (1000, 925, and 850 hPa) associated with the NPO during the boreal winter−spring (from November of year (−1) to May of year (0)), which are obtained by regressing U, V, and T anomalies at 1000, 925, and 850 hPa on the concurrent NPO index (Supplementary Fig. 16). Two model experiments were performed with the only difference being whether the positive NPO forcing is assimilated using the WCDA system and added to the climatological annual cycle of the atmosphere component of the FGOALS-g2 model. The experiment with positive NPO forcing is denoted as the NPO forcing experiment (EXP$_{NPO}$), whereas the experiment with the climatological annual cycle of the atmosphere component is denoted as the CTRL experiment (EXP$_{CTRL}$). The CTRL experiment was run for 100 years, with its atmospheric composition, solar irradiance, and land cover fixed at 1850 values. The NPO forcing experiment consisted of 60 member ensemble simulations from the CTRL, each initialized with conditions on 1 November from the last 60 years of the 100-year integration.

## Data availability

The HadISST dataset is available at http://www.metoffice.gov.uk/hadobs/hadsst3/. The ERSST dataset is available at https://psl.noaa.gov/data/gridded/data.noaa.ersst.v4.html. The Kaplan SST dataset is available at https://psl.noaa.gov/data/gridded/data.kaplan_sst.html. The COBE SST dataset is available at https://psl.noaa.gov/data/gridded/data.cobe2.html. The NCEP/NCAR monthly reanalysis is available at http://www.esrl.noaa.gov/psd/data/gridded/data.ncep.reanalysis.html. The HadSLP2 dataset is available at http://www.metoffice.gov.uk/hadobs/hadslp2/. The Niño3.4 index is provided by the Climate Prediction Center at https://psl.noaa.gov/data/correlation/nina34.anom.data. The CMIP5 simulation dataset is available at https://esgf-node.llnl.gov/projects/cmip5/, and the CMIP6 simulation dataset is available at https://esgf-node.llnl.gov/projects/cmip6/.

## Code availability

The data in this study were analyzed with NCAR Command Language (NCL; http://www.ncl.ucar.edu/). All relevant codes used in this study are available, upon request, from the corresponding author R.Q.D.

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

## Acknowledgements

This research was jointly supported by the National Natural Science Foundation of China (41790474, 41975070), and China's National Key Research and Development Projects (2020YFA0608402). Y.H.T. was supported by MOST Grant# 107-2611-M-002-013-MY4 "Improving the Decadal Climate Prediction using a New Fully Coupled Global Climate System". We thank Dr. Lijuan Li from Institute of Atmospheric Physics, Chinese Academy of Sciences for her support of the modelling experiments.

## Author contributions

R.Q.D. and J.P.L. designed the study. R.Q.D. wrote the paper. L.S. performed the data analysis and prepared all figures. F.F.L. conducted the modelling experiments. Y.-H.T., E.D., J.-Y.Y., C.Z.W., C.S., J.-J.L., K.-J.H., and Z.-Z.H. contributed to the interpretation of the results and the improvement of the manuscript.

## Competing interests

The authors declare no competing interests.
