## [Peer Review File · Nature Communications]

Multi-year El Niño events tied to the North Pacific OscillationReviewer #1 (Remarks to the Author):

In this paper the authors propose that the 5 prolonged El Niño events (also called multi-year El Niño events) observed since 1950 are associated to a two-way feedback mechanism between NPO forcing and the developed warm conditions of the first year of the event. NPO forcing the first year acts as an external forcing of the El Niño the first year, which in turn re-energize the NPO forcing for the second year through atmospheric teleconnection providing persistence to the equatorial warm conditions the second year through a pathway similar to that of the first year (i.e. NPO-Meridional mode-ENSO relationship). This two-way feedback mechanism is apparently less effective for the prolonged La Niña events based on the analysis of the 7 identified La Niña events. They test their hypothesis in the models participating to CMIP5/6 and also from a sensitivity experiment with the FGOALS-g2 model that is developed by LAGS, and show in particular a clear relationship between the increase in NPO variance and that of the number of multi-year El Niño events in the future climate, suggestive that more events of that kind are expected in the future.

These results have clear implications in terms of predicting ENSO impacts worldwide and are thus considered important.

While the paper builds upon material of various previous studies on the topic, it also proposes here an interesting perspective, which seems to be well supported by the comprehensive model and data analyses. I thus recommend publication.

Other comments

I. 66-68 : This statement suggests a causal relationship between multi-year El Niño and marine heat waves, which the proposed references do not really support.

I. 103-106 (Supplementary Figure 3): the map for 1986 FMA (c) does not evidence a clear North-South dipole in the North Pacific. In Figure f, explain how the dispersion on the mean bar is derived. We wonder what is the statistical significance of having an FMA NPO index above 1 during 5 multi-year El Niño events.

I. 109 : was it the monthly mean NPO index that was normalized of the FMA NPO index?

I. 113-117: The causal relationship between the two statement is not obvious. In particular there is a tendency of a La Niña state in the maps of Supplementary Figure 4

Supplementary Figure 5: It seems to indicate that the persistence of NINO34 is comparable to that of the lead-lag correlation between FMA NPO index and NINO34. Not sure this indicates that there is an influence of the NPO on the duration of El Niño. Could you clarify what this diagnostics is aimed at? In addition the authors argue that NPO contributes to a prolonged warming while the correlation is no longer statistically significant from JJA(1).

Reviewer #2 (Remarks to the Author):

This paper presents a novel theory to explain infrequent multi-year El Niño events. The study follows a series of studies explaining Pacific climate phenomenon as a result of a feedback interaction (occurring over multiple years and via PMM and ENSO teleconnections) between Tropical Pacific modes, dominated by ENSO, and North Pacific modes, which include Aleutian Low, NPO and NPGO, and PDO. All these studies, including the current paper, are based on the seminal work of Di Lorenzo et al. (2015), which was the first to explain aspects of the Pacific climate variability, specifically its low frequency component (i.e., Pacific decadal variability), assuming a feedback mechanism between ENSO and extratropical modes. Based on this theory, a 2016 study has explained the persistence in the North Pacific of a large SST anomaly that occurred

during 2014/2015. Now the authors use the same framework to explain the multi-year El Niño events as a feedback (mediated in the subtropics by the PMM) between NPO, CP/EP El Niño. The authors explored this hypothesis using observation and by performing simulations with an ocean-atmosphere coupled climate model. The manuscript, which is clear, well-written and with convincing arguments supported by well-posed modelling experiments, represents a good contribution to a phenomenon that has received only little attention from the scientific community. However, I have a two main comments and few suggestions I would like to the authors to address before the manuscript will be suitable for publication with Nature Communications.

Major comments:

The first year (JFM(0)) of the composite analysis presented in Figure 2 shows a pattern that might indicate a La Niña event co-occurring with a weak PMM. In contrast with a regular PMM, the cold SST anomaly in the equatorial Pacific seems to be stronger than the positive SST anomaly in the subtropics (for the PMM is the opposite.) However, the proposed explanation for the multi-year El Niño events does not include any relevant role for La Niña. Is it possible that this La Niña-like pattern plays a role in the multi-year El Niño event? How do you explain this cold SST anomaly in the equatorial Pacific?

In the last section, "Projections of future climate change", the authors notice that multi-year El Niño events have occurred more often during recent years and explore the possibility that these long-lasting events might become more frequent under anthropogenic forcing. Furthermore, they hypothesise that the increased frequency in multi-year events might be driven by enhanced NPO variance under climate change. This is an interesting hypothesis, and I appreciate their attempt find some evidence of this by looking at CMIP5/6 projection. However, I find this analysis quite preliminary and rather inconclusive. Even assuming all the El Niño events to be independent from each other, enhanced NPO (i.e., an ENSO precursor) variance would lead to more El Niño events, and thus the likelihood of occurrence of a multi-year event. This section is quite interesting and should be expanded a little.

In addition to further analysis supporting your point, I suggest to add in figure 6, a subplot comparing the occurrences of multi-year events between period s? with low and high radiative forcing (e.g., 1960-1990 vs 2060-2090).

Furthermore, I am a little surprised the authors did not mention the study of Liguori and Di Lorenzo (2018); it supports the authors' hypothesis of an enhanced NPO variance under anthropogenic forcing. Specifically, Liguori and Di Lorenzo analyse a large ensemble of climate simulations and found a robust increase in the PMM activity under anthropogenic forcing (i.e., RCP8.5). While they explained this increase with a theory involving the WES feedback, it is possible that the real cause is an increase in the NPO variance.

Liguori G. and Di Lorenzo E., 2018: Meridional Modes and Increasing Pacific decadal variability under greenhouse forcing. *Geophysical Research Letters*, 45.
<https://doi.org/10.1002/2017GL076548>.

Minor comments:

Figure 4 and the schematic used to explain the Pacific Decadal variability presented in the Di Lorenzo et al., 2015. are very similar. However, I find the iconic infographic of Di Lorenzo's study more informative (and attractive), but this might be just matter of personal taste.

Line 255: " indicates that 71% of multi-year El Niño events are preceded by positive NPO events" Isn't this statistic somewhat expected since NPO is an El Niño precursor? I think it is important to compare this number with what one would expect considering all NPO/ENSO event independent from each other (this would be the null hypothesis).

Line 324-325: "Our study differs from previous studies by attributing the dominant source of multi-year El Niño to NPO-like extratropical". Perhaps, you should soften this statement as I don't think you present enough evidence for the NPO to be a dominant source. As you write in line 361 there are many other sources.

Reviewer #3 (Remarks to the Author):

Comments on the manuscript entitled "Multi-year El Niño events tied to the North Pacific Oscillation" by R. Ding et al.

In this study the authors investigated the physical reasons for the multi-year El Niño events based on both data diagnoses and numerical experiments. It is found that the North Pacific oscillation (NPO) is a key source for such events. A two-way feedback mechanism between the tropical sea surface temperature (SST) anomalies and extratropical NPO was proposed to explain the long-lasting El Niños. This study is interesting and important in understanding the ENSO variability. However, there are major flaws existing in the present study. This manuscript might be accepted for publication after major revision. My specific comments on this manuscript are listed below.

Specific comments:

1. As revealed in previous and this studies, the NPO during the boreal winter can induce a central Pacific (CP) El Niño. However, in this study the Niño3.4 index was used in representing ENSO but not an index of CP El Niño. Why is not an index of CP El Niño used in this study? How about the results will be if a CP El Niño index is used? The authors need to give explanations on these issues.
2. In the section of Introduction, the authors mentioned that the "NPO may also play an important (but not dominant) role in developing multi-year La Niña events through a similar mechanism" (Lines 94–96). However, in the manuscript the multi-year La Niña events were almost not dealt with although they were listed in Supplementary Table 1. I suggest the authors give evidence for such statement.
3. It seems that the authors chose the multi-year El Niño events subjectively based on Supplementary Fig. 1. I suggest using an objective method to define and choose objectively the multi-year El Niño events.
4. It is hard to see that the SLP anomalies in Supplementary Figs. 3a-3e were similar to a typical NPO. The authors should not only show the NPO indices (Fig. 1a), but also calculate the spatial correlations between the SLP anomalies shown in Figs. 3a-3e and those associated with a typical NPO to illustrate how similar the SLP anomalies to those of the NPO.
5. Based on Supplementary Fig. 4, the authors concluded that "the NPO signatures preceding multi-year El Niño events were not accompanied simultaneously by large sea surface temperature (SST) anomalies in the tropical Pacific" (Lines 113-117). Such conclusion may not be correct. As seen in Supplementary Fig. 4, in the tropical Pacific a La Niña type of SST occurred obviously in Supplementary Figs. 4b-4e. The authors should provide discussions on such feature. In fact, Fig. 2a also shows a clear feature of La Niña.
6. Supplementary Fig. 5 shows that the positive correlation of JFM(0) NPO index with the lagged Niño3.4 SST persisted through the following year until JFM(2). Such long-lasting correlation should be effective for all El Niño events. Physical explanations should be given why some El Niño events are long-lasting ones but some not in the condition of the long-lasting effect of NPO.
7. Based on Fig. 2, the authors gave a conclusion that "positive SST anomalies in JFM(0) extended from the subtropical northeastern Pacific to the central equatorial Pacific and led to equatorial Pacific warming during the subsequent JFM(1)". However, as seen in Supplementary Fig. 6, in JAS(0) a weak warming occurred in the equatorial eastern

Pacific and strengthened afterwards, which does not seem to be related to the SST anomalies in the subtropical northeastern Pacific but developed locally. The authors should provide evidence to prove if the positive SST anomalies in the central and eastern equatorial Pacific came from the subtropical northeastern Pacific or from local development.

8. I suggest adding surface winds in all the figures showing SLP anomalies in Supplementary Fig. 11 to see if the circulation anomalies associated with the negative SLP anomalies around Hawaii weakened the trade winds in the equatorial Pacific.

9. In this study the Pacific meridional mode (PMM) is mentioned. I suggest using the North Pacific meridional mode (NPMM) instead of PMM in the manuscript because in the South Pacific Ocean there exists another PMM, the South Pacific meridional mode (SPMM), which is also closely related with ENSO (Min et al., 2017, *J. Climate*, 30, 1705–1720).

10. The historical simulations of CMIP5/6 models were from 1900 to 1999, during which the three multi-year El Niño events of 1957/58/59, 1968/69/70, and 1986/87/88 were included. I suggest the authors compare the three events in simulations with those in observations to see how well the simulated ones in representing the observed ones.

11. In this study the role played by NPO in multi-year El Niño events were stressed. In fact, the effect from the subtropical southern Pacific should also take effect in the multi-year El Niño events. For example, the evolution of the long-lasting El Niño events in 2014/15/16 were affected by the system in the subtropical southern Pacific (Min et al., 2015, *Geophys. Res. Lett.*, 42, 6762–6770; Su et al., 2018, *J. Climate*, 31, 877–893). I suggest adding necessary discussions in the manuscript by referring these previous studies.

**"Multi-year El Niño events tied to the North Pacific Oscillation"
(NCOMMS-21-46528A) by Ding et al.
Responses to Reviewers #1, #2 and #3**

March 8, 2022

I. Reviewer #1

I-A. Response to general comments

In this paper the authors propose that the 5 prolonged El Niño events (also called multi-year El Niño events) observed since 1950 are associated to a two-way feedback mechanism between NPO forcing and the developed warm conditions of the first year of the event. NPO forcing the first year acts as an external forcing of the El Niño the first year; which in turn re-energize the NPO forcing for the second year through atmospheric teleconnection providing persistence to the equatorial warm conditions the second year through a pathway similar to that of the first year (i.e. NPO-Meridional mode-ENSO relationship). This two-way feedback mechanism is apparently less effective for the prolonged La Niña events based on the analysis of the 7 identified La Niña events. They test their hypothesis in the models participating to CMIP5/6 and also from a sensitivity experiment with the FGOALS-g2 model that is developed by LAGS, and show in particular a clear relationship between the increase in NPO variance and that of the number of multi-year El Niño events in the future climate, suggestive that more events of that kind are expected in the future.

These results have clear implications in terms of predicting ENSO impacts worldwide and are thus considered important. While the paper builds upon material of various previous studies on the topic, it also proposes here an interesting perspective, which seems to be well supported by the comprehensive model and data analyses. I thus recommend publication.

Response: We sincerely thank the reviewer for his/her positive comments and

thoughtful review, which are a big help to the authors for improving this manuscript. We have carefully revised the manuscript according to the comments and suggestions raised by the reviewer. Please find below a detailed point-by-point response to all comments (reviewers' comments in black, our replies in blue).

I-B: Response to other comments

1. 66-68 : *This statement suggests a causal relationship between multi-year El Nino and marine heat waves, which the proposed references do not really support.*

Response: We thank the reviewer for pointing this out. We have rewritten the sentence by removing marine heat waves and changing the references accordingly.

2. 103-106 (Supplementary Figure 3): *the map for 1986 FMA (c) does not evidence a clear North-South dipole in the North Pacific.*

Response: We appreciate the helpful comment from the reviewer. We have redrawn former Supplementary Figure 3 and extended the figure northward to 80°N (the original map was only to 60°N). In the modified plot, the SLP map in JFM 1986 generally shows a north-south dipole pattern in the North Pacific, although the northern pole is less evident than the southern pole (please see Figure A1c below).

In addition, to check how well the preceding JFM(0) SLP patterns of the five multi-year El Niño events match the NPO typical pattern, we have calculated the spatial correlation coefficients of the JFM(0) SLP pattern with the typical NPO pattern (Figure A1f; please also see our response to Reviewer #3 Specific comments #4). The spatial correlation coefficients for the five multi-year El Niño events are all significant above the 95% significance level (Figure A1h). The spatial correlation coefficient for the 1986 map is as high as 0.67. These results

support our argument that the North Pacific SLP precursor patterns for multi-year El Niño events are similar to a typical NPO pattern.

Figure A1. The JFM(0) SLP anomalies prior to multi-year El Niño events in observations. (a–e) The JFM(0) SLP anomalies for the five observed multi-year El Niño events, respectively. **(f)** Regressions of SLP anomalies onto the normalized JFM NPO index. **(g)** The normalized JFM(0) NPO index for the five observed multi-year El Niño events. The horizontal dashed line represents one positive standard deviation. **(h)** The spatial correlation coefficients of the five JFM(0) SLP patterns in (a–e) with the typical NPO pattern in (f) over the North Pacific region (15° – 70° N and 150° E– 120° W). The horizontal dashed line shows the 95% confidence level. The confidence level at which the spatial correlations are significant is calculated using an effective number of spatial degrees of freedom. The effective numbers of spatial degrees of freedom for the five JFM(0)

SLP patterns are slightly different, and we have taken the minimum value.

In Figure f, explain how the dispersion on the mean bar is derived. We wonder what is the statistical significance of having an FMA NPO index above 1 during 5 multi-year El Niño events.

Response: The mean bar in Figure 1a and former Supplementary Figure 3f was obtained by computing the composite (five events mean) NPO indices. The statistical significance of the composite anomalies (i.e., the dispersion on the mean bar) was evaluated based on a two-tailed Student's t -test. The formula for computing the t -value and degrees of freedom for paired t -test is:

$$t = \frac{\bar{X}_1 - \bar{X}_2}{\sqrt{\frac{(n_1 - 1)S_1^2 + (n_2 - 1)S_2^2}{n_1 + n_2 - 2} \left(\frac{1}{n_1} + \frac{1}{n_2} \right)}}$$

where S_1 and S_2 are the variances of variables X_1 (the NPO index of multi-year El Niño events) and X_2 (the NPO index of all winters), respectively, and n_1 and n_2 are the numbers of variables X_1 and X_2 , respectively. According to the above equation, we can obtain:

$$\bar{X}_1 - \bar{X}_2 = t \sqrt{\frac{(n_1 - 1)S_1^2 + (n_2 - 1)S_2^2}{n_1 + n_2 - 2} \left(\frac{1}{n_1} + \frac{1}{n_2} \right)}$$

According to the table of t -distribution table, we can obtain the t value significant at the 95% level. S_1 , S_2 , n_1 , and n_2 are known in advance. Therefore, according to the above equation, we can obtain the 95% confidence level of the average NPO indices over the five events (i.e., the error bar).

We would like to clarify that we only provide the statistical significance of the average NPO index (i.e., error bars) for the five events, but not for the JFM NPO index above 1. The horizontal dashed line in Figure 1a of the main manuscript simply shows that the JFM(0) NPO index for all 5 multi-year El Niño events exceeds 1.0 standard deviation.

3. *109: was it the monthly mean NPO index that was normalized of the FMA NPO index?*

Response: Yes, we first obtain the time series of the JFM(0) average NPO index and then normalize it with its standard deviation.

4. *113-117: The causal relationship between the two statement is not obvious. In particular there is a tendency of a La Nina state in the maps of Supplementary Figure 4.*

Response: Thanks for the insightful comments from the reviewer. We agree with the reviewer that there are some inaccuracies with our statements here. The sentence “these NPO signatures preceding multi-year El Niño events are not accompanied simultaneously by large sea surface temperature (SST) anomalies in the tropical Pacific, implying that they may originate primarily from intrinsic atmospheric variability in the North Pacific rather than from tropical SST forcing” has been removed from the revised manuscript. Instead, we have added the following sentences in the section “Summary and Discussion” of the revised manuscript (please see lines 331-341):

“While the North Pacific precursor patterns for multi-year El Niño events have a strong NPO/NPMM structure, we note that in observations there is also an evident La Niña-like condition in the precursor patterns (Fig. 2a). Given the few realizations in the observational record, the dynamical significance of the tropical Pacific cooling is still unclear. In the CMIP5/6 models, the probability of NPO events co-occurring with La Niña prior to multi-year El Niño events is relatively low (~10%; Fig. 1e). The multi-model ensemble precursor patterns also exhibit a weak cooling in the tropical Pacific, but its amplitude is much smaller compared to the strength of the NPO signal (Supplementary Fig. 15a). In this regard, further modelling studies are required to explore the relative

importance of the NPO and La Niña states in developing subsequent multi-year El Niño events.”

5. *Supplementary Figure 5: It seems to indicate that the persistence of NINO34 is comparable to that of the lead-lag correlation between FMA NPO index and NINO34. Not sure this indicates that there is an influence of the NPO on the duration of El Nino. Could you clarify what this diagnostics is aimed at? In addition the authors argue that NPO contributes to a prolonged warming while the correlation is no longer statistically significant from JJA(1).*

Response: We are grateful for the reviewer’s helpful comments. Former Supplementary Figure 5 shows that the correlations between the NPO and Niño3.4 indices remain positive through the following year until JFM(2), suggesting that the NPO-related El Niño state may persist for more than two years. Based on these persistent positive correlations, we argue that the NPO tends to induce a slower phase transition of El Niño, thereby creating favorable conditions for the occurrence of multi-year El Niño events.

However, as the reviewer mentioned, although the correlations between the NPO and Niño3.4 indices remain positive, they are no longer statistically significant from JJA(1). We speculated that there are two reasons why the correlations between the NPO and Niño3.4 indices are no longer significant after JJA(1). Firstly, the NPO is likely to be followed by not only multi-year El Niño but also single-year El Niño. Secondly, the number of multiyear El Niño events preceded by the NPO is limited in the observations. In the light of the reviewer’s comments, we decided to remove the former Supplementary Figure 5 from the revised version since it did not add more information for the present article. The paragraph containing the former Supplementary Figure 5 has also been removed from the revised manuscript.

II. Reviewer #2

II-A. Response to general comments

This paper presents a novel theory to explain infrequent multi-year El Niño events. The study follows a series of studies explaining Pacific climate phenomenon as a result of a feedback interaction (occurring over multiple years and via PMM and ENSO teleconnections) between Tropical Pacific modes, dominated by ENSO, and North Pacific modes, which include Aleutian Low, NPO and NPGO, and PDO. All these studies, including the current paper, are based on the seminal work of Di Lorenzo et al. (2015), which was the first to explain aspects of the Pacific climate variability, specifically its low frequency component (i.e., Pacific decadal variability), assuming a feedback mechanism between ENSO and extratropical modes. Based on this theory, a 2016 study has explained the persistence in the North Pacific of a large SST anomaly that occurred during 2014/2015. Now the authors use the same framework to explain the multi-year El Niño events as a feedback (mediated in the subtropics by the PMM) between NPO, CP/EP El Niño. The authors explored this hypothesis using observation and by performing simulations with an ocean-atmosphere coupled climate model.

The manuscript, which is clear, well-written and with convincing arguments supported by well-posed modelling experiments, represents a good contribution to a phenomenon that has received only little attention from the scientific community. However, I have a two main comments and few suggestions I would like to the authors to address before the manuscript will be suitable for publication with Nature Communications

Response: We appreciate all the thoughtful comments and constructive suggestions from the reviewer. We have carefully modified the manuscript according to the comments and suggestions raised by the reviewer. Please see our point-by-point responses (in blue) to the comments raised by the reviewer below.

II-B: Response to major comments

1. *The first year (JFM(0)) of the composite analysis presented in Figure 2 shows a pattern that might indicate a La Niña event co-occurring with a weak PMM. In contrast with a regular PMM, the cold SST anomaly in the equatorial Pacific seems to be stronger than the positive SST anomaly in the subtropics (for the PMM is the opposite.) However, the proposed explanation for the multi-year El Niño events does not include any relevant role for La Niña. Is it possible that this La Niña-like pattern plays a role in the multi-year El Niño event? How do you explain this cold SST anomaly in the equatorial Pacific?*

Response: Thanks for the insightful comments from the reviewer. As the reviewer mentioned, positive SST anomalies in the subtropical northeastern Pacific are accompanied simultaneously by negative SST anomalies in the eastern equatorial Pacific in the first year (JFM(0)) of the composite analysis presented in Figure 2. On the one hand, these negative SST anomalies may be partly forced by the NPO, which can simultaneously induce both positive SST anomalies in the subtropical northeastern Pacific and negative SST anomalies in the eastern equatorial Pacific (i.e., the PMM-like pattern). On the other hand, they may be a result of La Niña events co-occurring with positive NPO events as the reviewer suggested. For example, the positive NPO event in JFM 2018 co-occurs with a weak La Niña event in the tropical Pacific. The occurrence of La Niña events would further strengthen the NPO-induced negative SST anomalies in the eastern equatorial Pacific. Considering that negative SST anomalies concentrate in the eastern equatorial Pacific rather than in the central equatorial Pacific, they do not seem to have a significant direct impact on the concurrent NPO and subsequent NPO-PMM-ENSO relationship. Nevertheless, we cannot rule out the possibility that the preceding tropical Pacific cooling may also contribute to the occurrence of multi-year El Niño events through coupled dynamics of the tropical ocean-atmosphere system, which deserves further in-depth study.

We have added a discussion of the tropical cooling in the section “Summary and Discussion” of the revised manuscript (please see lines 331-341):

“While the North Pacific precursor patterns for multi-year El Niño events have a strong NPO/NPMM structure, we note that in observations there is also an

evident La Niña-like condition in the precursor patterns (Fig. 2a). Given the few realizations in the observational record, the dynamical significance of the tropical Pacific cooling is still unclear. In the CMIP5/6 models, the probability of NPO events co-occurring with La Niña prior to multi-year El Niño events is relatively low (~10%; Fig. 1e). The multi-model ensemble precursor patterns also exhibit a weak cooling in the tropical Pacific, but its amplitude is much smaller compared to the strength of the NPO signal (Supplementary Fig. 15a). In this regard, further modelling studies are required to explore the relative importance of the NPO and La Niña states in developing subsequent multi-year El Niño events.”

2. *In the last section, "Projections of future climate change", the authors notice that multiyear El Niño events have occurred more often during recent years and explore the possibility that these long-lasting events might become more frequent under anthropogenic forcing. Furthermore, they hypothesise that the increased frequency in multi-year events might be driven by enhanced NPO variance under climate change. This is an interesting hypothesis, and I appreciate their attempt find some evidence of this by looking at CMIP5/6 projection. However, I find this analysis quite preliminary and rather inconclusive. Even assuming all the El Niño events to be independent from each other, enhanced NPO (i.e., an ENSO precursor) variance would lead to more El Niño events, and thus the likelihood of occurrence of a multi-year event. This section is quite interesting and should be expanded a little.*

Response: Thanks for the constructive comments from the reviewer. Motivated by the reviewer’s comment, we have now expanded this section to include not only the inter-model relationship between the change in the number of multi-year El Niño events and the change in the NPO variance (see Figure B1c below), but also the inter-model relationship between the change in the ratio of the number of multi-year El Niño events to the total number of El Niño events and the change in the NPO variance (see Figure B1d below). We note that models with the increased variance of the NPO index in the future climate systematically produce a more frequent occurrence of multi-year El Niño events, and this tendency is statistically significant. Furthermore, the ratio of the number of multi-year El

Niño events to the total number of El Niño events is also projected to increase due to increased NPO variance, regardless of how the total number of El Niño events changes in response to anthropogenic global warming. These results support our hypothesis that the increased frequency in multi-year El Niño events might be driven by enhanced NPO variance under climate change.

It should also be pointed out that conclusively demonstrating that the increase in NPO variance is leading to more frequent multi-year El Niño events is beyond the scope of this paper, which focuses on establishing the evidence that NPO dynamics act as a driver for multi-year El Niño events. The multi-model ensemble analysis is reported as a suggestion for further studies. We have changed the last sentence from:

“This implies that if the NPO variance increases in a warmer climate, multi-year El Niño events will occur more frequently.”

to

“Although further studies are required to determine how the NPO dynamics are responding to a warmer climate, these results suggest that an increase in NPO variance might not only energize the ENSO system (e.g. Liguori and Di Lorenzo, 2018) but also might lead to more frequent multi-year El Niño events.”

Liguori G. and Di Lorenzo E. Meridional Modes and Increasing Pacific decadal variability under greenhouse forcing. *Geophys. Res. Lett.*, 45 (2018).

In addition to further analysis supporting your point, I suggest to add in figure 6, a subplot comparing the occurrences of multi-year events between periods with low and high radiative forcing (e.g., 1960-1990 vs 2060-2090).

Response: We appreciate the reviewer's suggestions. In the light of the reviewer's suggestions, we have now added a subplot in the former Figure 6 comparing the number of multi-year El Niño events that occurred in the present (1900–1999)

and future (2000–2099) climates using the 23 selected CMIP5/6 models (see Figure B1a below). The present and future warmer climates are represented by the historical (HIST) and Representative Concentration Pathway 8.5 (RCP8.5) experiments, respectively. 16 of the 23 models (70%) simulate an increased frequency of multi-year El Niño events in the future warmer climate, with a good inter-model consensus (see Figure B1a below). The ensemble mean increase is 43% (from about 7 events per 100 years in the present climate to 10 events per 100 years in the future climate), which is significant above the 95% confidence level according to a bootstrap test (Figure B1b).

Furthermore, I am a little surprised the authors did not mention the study of Liguori and Di Lorenzo (2018); it supports the authors' hypothesis of an enhanced NPO variance under anthropogenic forcing. Specifically, Liguori and Di Lorenzo analyse a large ensemble of climate simulations and found a robust increase in the PMM activity under anthropogenic forcing (i.e., RCP8.5). While they explained this increase with a theory involving the WES feedback, it is possible that the real cause is an increase in the NPO variance.

Response: We thank the reviewer for pointing this out. We have now added the following sentence in the section “Projections of future climate change” of the revised manuscript (please see lines 300-302):

“Previous studies have suggested that the NPO variance may increase in a warmer climate either due to a more energetic WES feedback (e.g. Liguori and Di Lorenzo, 2018) or due to a stronger NPO excitation driven by the Kuroshio Extension downstream atmospheric response (Joh and Di Lorenzo, 2019; Joh et al. 2021)”.

Liguori G. and Di Lorenzo E. (2018) Meridional Modes and Increasing Pacific decadal variability under greenhouse forcing. *Geophys. Res. Lett.*, 45.

Joh, Y., and E. Di Lorenzo (2019), Interactions between Kuroshio Extension and Central

Tropical Pacific lead to preferred decadal-timescale oscillations in Pacific climate, *Sci. Rep.*, 9, 12.

Joh, Y., E. Di Lorenzo, L. Siqueira, and B. P. Kirtman (2021), Enhanced interactions of Kuroshio Extension with tropical Pacific in a changing climate, *Sci Rep*, 11(1), 12.

Figure B1. Relationship between future change in multi-year El Niño event frequency and NPO variance. (a) The number of multi-year El Niño events that

occurred in the present (1900 to 1999, blue) and future (2000 to 2099, red) climates. The seven models that simulate a decrease in number are grayed out. Error bars in the multi-model mean correspond to the 95% confidence interval. **(b)** Histograms of 10,000 realizations of a bootstrap method for the number of multi-year El Niño events in the present (blue) and future (red) climates. The blue and red lines indicate the mean values of the 10,000 realizations for the present and future climates, respectively. The gray shaded areas refer to the respective 1.0 SD of the 10,000 realizations. **(c)** Inter-model relationship between the change in the number of multi-year El Niño events and the change in the NPO variance. The linear regression line is shown by the solid line, with the significant correlation coefficient (R) and significance level (P) being indicated in the right bottom corner of the figure. **(d)** As in (c), but for the inter-model relationship between the change in the ratio of the number of multi-year El Niño events to the total number of El Niño events and the change in the NPO variance. The change in the number of multi-year El Niño events is defined as $\Delta=(m_2-m_1)/m_1$, where m_1 and m_2 represent the numbers of multi-year El Niño events that occurred in the present and future climates, respectively. The change in the ratio of the number of multi-year El Niño events to the total number of El Niño events is defined as $\Delta=m_2/n_2-m_1/n_1$, where n_1 and n_2 represent the total number of El Niño events that occurred in the present and future climates, respectively.

II-B: Response to minor comments

1. *Figure 4 and the schematic used to explain the Pacific Decadal variability presented in the Di Lorenzo et al., 2015. are very similar. However, I find the iconic infographic of Di Lorenzo's study more informative (and attractive), but this might be just matter of personal taste.*

Response: Our schematic used to explain the dynamics underlying the link between the NPO to multi-year ENSO events was motivated by Fig. 3 in Di Lorenzo et al (2015). The difference is that the schematic of Di Lorenzo et al (2015) aims to explain Pacific decadal variability (timescale > 8 years), while our study focuses on establishing the evidence that NPO dynamics act as a driver for multi-year El Niño events (dominated primarily by interannual variations in the 3 to 7 yr ranges). From this perspective, our schematic is a complement to and an extension of Di Lorenzo et al. (2015)'s schematic, and we argued that the two-way feedback between the tropics and extratropics may contribute to the occurrence of multi-year El Niño.

2. *Line 255: " indicates that 71% of multi-year El Niño events are preceded by positive NPO events" Isn't this statistic somewhat expected since NPO is an El Niño precursor? I think it is important to compare this number with what one would expect considering all NPO/ENSO event independent from each other (this would be the null hypothesis).*

Response: We appreciate the good suggestion from the reviewer. Following the reviewer's suggestion, we compared the probability of the NPO-preceded multi-year El Niño events with the probability of the NPO-preceded El Niño events (including both single-year and multi-year El Niño) and the probability of the NPO-preceded single-year El Niño events. The probability of the NPO-preceded multi-year El Niño events (71%) is significantly higher than the probability of the NPO-preceded El Niño events (41%) and the probability of the NPO-preceded single-year El Niño events (27%) (see Figure B2 below). These results support our view that although the NPO acts as a precursor for ENSO, multi-year El Niño events are more closely tied to the precedent NPO than single-year El Niño events. Figure B2 and the following sentences have been added to the section "Simulated impacts of the NPO on multi-year El Niño events" of the revised manuscript (please see Supplementary Fig. 14 and lines 247-252):

"The probability of the NPO-preceded multi-year El Niño events (71%) is significantly higher than the probability of the NPO-preceded El Niño events (including both single-year and multi-year El Niño) (41%) and the probability of the NPO-preceded single-year El Niño events (27%) (Supplementary 14), indicating that in contrast with single-year El Niño, multi-year El Niño is more closely tied to the precedent NPO."

Figure B2. The frequency ratio of the NPO-preceded El Niño/single-year El Niño events in the CMIP5/6 models. (a) The ratios of the NPO-preceded multi-year El Niño (red) and El Niño (including both single-year and multi-year El Niño, blue) events. Error bars in the multi-model mean indicate the 95% confidence interval. (b) As in (a) but for the ratios of the NPO-preceded multi-year El Niño (red) and single-year El Niño (blue) events. (c) Histograms of 10,000 realizations of a bootstrap method for the ratios of the NPO-preceded multi-year El Niño (red) and El Niño (blue) events. The red and blue lines indicate the mean values of the 10,000 realizations for the ratios of the NPO-preceded multi-year El Niño and El Niño events, respectively. The gray shaded areas refer to the respective 1.0 SD of the 10,000 realizations. (d) As in (c) but for histograms of 10,000 realizations of a bootstrap method for the ratios of the NPO-preceded multi-year El Niño (red) and single-year El Niño (blue) events.

3. Line 324-325: "Our study differs from previous studies by attributing the dominant source of multi-year El Niño to NPO-like extratropical". Perhaps, you should soften this statement as I don't think you present enough evidence for the

NPO to be a dominant source. As you write in line 361 there are many other sources.

Response: Following the reviewer's suggestion, we have rewritten the statement as follows:

“Our results differ from previous studies, and attribute **one of the dominant sources** of multi-year El Niño to NPO-like extratropical atmospheric variability rather than tropical ocean-atmosphere coupled variability”.

III. Reviewer #3

III-A. Response to general comments

In this study the authors investigated the physical reasons for the multi-year El Niño events based on both data diagnoses and numerical experiments. It is found that the North Pacific oscillation (NPO) is a key source for such events. A two-way feedback mechanism between the tropical sea surface temperature (SST) anomalies and extratropical NPO was proposed to explain the long-lasting El Niños. This study is interesting and important in understanding the ENSO variability. However, there are major flaws existing in the present study. This manuscript might be accepted for publication after major revision. My specific comments on this manuscript are listed below.

Response: We would like to thank the reviewer for his/her insightful and constructive comments and suggestions for the manuscript. The manuscript has been revised as per the comments given by the reviewer, and please see our point-by-point responses (in blue) to all the comments below.

III-B: Response to specific comments

1. *As revealed in previous and this studies, the NPO during the boreal winter can induce a central Pacific (CP) El Niño. However, in this study the Niño3.4 index was used in representing ENSO but not an index of CP El Niño. Why is not an index of CP El Niño used in this study? How about the results will be if a CP El Niño index is used? The authors need to give explanations on these issues.*

Response: Thanks for the constructive comments from the reviewer. According to the schematic diagram in Figure 4 of the manuscript, the boreal winter NPO induces a CP-type El Niño event over the equatorial Pacific during the subsequent winter through its effect on the NPM. This CP-type El Niño in turn feeds back into the North Pacific to force changes in atmospheric circulation over the Hawaiian region, which re-activate the NPM to favor the development of another CP-type or EP-type El Niño event. In this two-way feedback process

between the tropics and extratropics, the El Niño event of the first year is usually a CP type, but the El Niño event of the second year is not necessarily a CP type, and it could be an EP type. Therefore, if a CP El Niño index was used to measure the evolutions of multi-year El Niño events, it cannot well capture the temporal evolution characteristics of multi-year El Niño events. Taking the 2014/15/16 event as an example, the El Niño event during JFM of 2016 emerges as an EP type, and the CP El Niño index cannot capture the realistic intensity of this 2015/2016 El Niño event (see Figure C1 below). In contrast, as a commonly used index to define ENSO events, the Niño34 index is closely related to both indices of CP-type and EP-type ENSO, and was therefore used to represent ENSO in our study.

Figure C1. SST anomalies during boreal winter (JFM) of 2015 (a) and 2016 (b). The CP El Niño and Niño34 indices are provided in the top right.

2. *In the section of Introduction, the authors mentioned that the “NPO may also play an important (but not dominant) role in developing multi-year La Niña events through a similar mechanism” (Lines 94-96). However, in the manuscript the multi-year La Niña events were almost not dealt with although they were listed in Supplementary Table 1. I suggest the authors give evidence for such statement.*

Response: We appreciate the helpful comments from the reviewer. Our study focuses on establishing the evidence that NPO dynamics act as a driver for

multi-year El Niño events. Due to the length constraints of this paper, we placed the analysis of multi-year La Niña events in the section “Summary and discussion”. Nevertheless, our results provide compelling evidence that shows that the NPO may also play an important (but not dominant) role in developing multi-year La Niña events.

Firstly, we find that there is a relatively high likelihood of multi-year La Niña events in observations (57%) and CMIP5/6 models (77%) that are preceded by strong El Niño events in the tropical Pacific. In contrast, only 21% of multi-year La Niña events are preceded by negative NPO events alone without an accompanying strong El Niño in models. Given a relatively small percentage of the NPO-preceded multi-year La Niña events, we argue that the duration of La Niña events might be strongly affected by the strength of preceding El Niño events, and the NPO might not play a dominant role in developing multi-year La Niña events.

Secondly, the composite analysis of multi-year La Niña events preceded by the negative NPO event alone in CMIP5/6 models shows that a negative NPO event alone during JFM(0) can also lead to multi-year La Niña events through the two-way feedback mechanism between the tropics and extratropics. This suggests that although the phase transitions of La Niña are determined largely by the recharge process of equatorial oceanic heat content, the NPO may also play a role in developing multi-year La Niña events, resembling its role in developing multi-year El Niño events.

Based on the above two pieces of evidence, we argue that the NPO may also play an important (but not dominant) role in developing multi-year La Niña events. We agree with the reviewer that the current analysis of multi-year La Niña events is preliminary. We plan to conduct further systematic and in-depth investigations of multi-year La Niña events in future work. In the light of the reviewer’s

comments, we have removed the following sentence from the section “Introduction” of the revised manuscript:

“In addition, we argue that the NPO may also play an important (but not dominant) role in developing multi-year La Niña events through a similar mechanism.”

3. *It seems that the authors chose the multi-year El Niño events subjectively based on Supplementary Fig. 1. I suggest using an objective method to define and choose objectively the multi-year El Niño events.*

Response: We would like to clarify that we defined the multi-year El Niño/ La Niña events based on the following objective method (please see Methods for the definition of multi-year El Niño and La Niña events of the manuscript):

“For the observational analysis, multi-year El Niño and La Niña events are defined based on monthly SST anomalies averaged over the Niño3.4 region (5°S–5°N, 120°–170°W; termed the “Niño3.4 index”) for 1950–2020. The monthly Niño3.4 index is linearly detrended and smoothed with a 3-month-running-mean filter. We denote the year when El Niño and La Niña first develop as year (0), and the following two years as year (1) and year (2), respectively. A multi-year El Niño (La Niña) event is identified when the Niño3.4 index is over 0.75 (below –0.75) standard deviations in any month during October(0) to February(1), and remains above zero thereafter and exceeds 0.5 (–0.5) standard deviations again in any month during October(1) to February(2) (Wu et al. 2019). According to these criteria, there are five multi-year El Niño events and seven multi-year La Niña events for 1950–2020 (Supplementary Table 1).”

Wu, X., Okumura, Y. M., & DiNezio, P. N. What controls the duration of El Niño and La

4. *It is hard to see that the SLP anomalies in Supplementary Figs. 3a-3e were similar to a typical NPO. The authors should not only show the NPO indices (Fig. 1a), but also calculate the spatial correlations between the SLP anomalies shown in Figs. 3a-3e and those associated with a typical NPO to illustrate how similar the SLP anomalies to those of the NPO.*

Response: Thanks for the constructive suggestion from the reviewer. Following the reviewer’s suggestion, we have calculated the spatial correlation coefficients of the JFM(0) SLP pattern with the typical NPO pattern. The spatial correlation coefficients for the five multi-year El Niño events are all significant above the 95% significance level (please see Supplementary Fig. 4 and lines 107-110 of the revised manuscript). These results support our argument that the JFM(0) North Pacific SLP anomaly patterns for multi-year El Niño events are similar to a typical NPO pattern.

5. *Based on Supplementary Fig. 4, the authors concluded that “the NPO signatures preceding multi-year El Niño events were not accompanied simultaneously by large sea surface temperature (SST) anomalies in the tropical Pacific” (Lines 113-117). Such conclusion may not be correct. As seen in Supplementary Fig. 4, in the tropical Pacific a La Niña type of SST occurred obviously in Supplementary Figs. 4b-4e. The authors should provide discussions on such feature. In fact, Fig. 2a also shows a clear feature of La Niña.*

Response: We thank the reviewer for pointing this out. We agree with the reviewer that our wording was imprecise. The sentence “the NPO signatures preceding multi-year El Niño events were not accompanied simultaneously by large sea surface temperature (SST) anomalies in the tropical Pacific” has been deleted from the revised manuscript. Following the reviewer’s suggestion, we have added a discussion of the preceding tropical Pacific cooling in the section “Summary and Discussion” of the revised manuscript (please see lines 331-341):

“While the North Pacific precursor patterns for multi-year El Niño events have a strong NPO/NPMM structure, we note that in observations there is also an evident La Niña-like condition in the precursor patterns (Fig. 2a). Given the few realizations in the observational record, the dynamical significance of the tropical Pacific cooling is still unclear. In the CMIP5/6 models, the probability of NPO events co-occurring with La Niña prior to multi-year El Niño events is relatively low (~10%; Fig. 1e). The multi-model ensemble precursor patterns also exhibit a weak cooling in the tropical Pacific, but its amplitude is much smaller compared to the strength of the NPO signal (Supplementary Fig. 15a). In this regard, further modelling studies are required to explore the relative importance of the NPO and La Niña states in developing subsequent multi-year El Niño events.”

6. *Supplementary Fig. 5 shows that the positive correlation of JFM(0) NPO index with the lagged Niño3.4 SST persisted through the following year until JFM(2). Such long lasting correlation should be effective for all El Niño events. Physical explanations should be given why some El Niño events are long-lasting ones but some not in the condition of the long-lasting effect of NPO.*

Response: Thanks for the important comments. According to the dynamical hypothesis that we have proposed (please see the schematic diagram in Figure 4 of the manuscript), both the NPO and its induced CP El Niño are necessary for generating multi-year El Niño events. If the JFM(0) NPO is followed by a CP El Niño in the following JFM(1), this CP El Niño will likely re-intensify in the subsequent winter through the two-way feedback mechanism between the tropics and extratropics, lasting for two years or longer and finally resulting in multi-year El Niño events. However, previous studies have suggested that the NPO tends to induce the CP-type El Niño, but not in all cases (Ding et al. 2015). Equatorial ocean dynamics, such as zonal advection in the tropical Pacific, can extend NPO-induced warming in the central equatorial Pacific eastwards, leading to EP El Niño events (Pegion et al. 2013). If the JFM(0) NPO is followed by an EP El

Niño in the following JFM(1), then this EP El Niño is likely to evolve rapidly to La Niña-like conditions in the subsequent winter, finally leading to a single-year El Niño event. Therefore, it is important to consider the pattern of the NPO-induced El Niño when judging whether this El Niño event is a long-lasting one or not.

Ding, R., Li, J., Tseng, Y. H., Sun, C., & Guo, Y. The Victoria mode in the North Pacific linking extratropical sea level pressure variations to ENSO. *J. Geophys. Res. Atmos.* 120, 27–45 (2015).

Pegion, K., & Alexander, M. The seasonal footprinting mechanism in CFSv2: simulation and impact on ENSO prediction. *Clim. Dyn.* 41, 1671–1683 (2013).

7. *Based on Fig. 2, the authors gave a conclusion that “positive SST anomalies in JFM(0) extended from the subtropical northeastern Pacific to the central equatorial Pacific and led to equatorial Pacific warming during the subsequent JFM(1)”. However, as seen in Supplementary Fig. 6, in JAS(0) a weak warming occurred in the equatorial eastern Pacific and strengthened afterwards, which does not seem to be related to the SST anomalies in the subtropical northeastern Pacific but developed locally. The authors should provide evidence to prove if the positive SST anomalies in the central and eastern equatorial Pacific came from the subtropical northeastern Pacific or from local development.*

Response: We appreciate the helpful comments from the reviewer. We note in Supplementary Figure 4 of the revised manuscript (i.e., former Supplementary Fig. 6) that anomalous easterlies simultaneously occur in the equatorial eastern Pacific during JAS(0), which are unfavorable for the warming in the equatorial eastern Pacific by enhanced upwelling. It appears that the equatorial eastern Pacific warming may not arise from the local interactions between SST and surface winds. The weak warming in the equatorial eastern Pacific is more likely to arise from anomalous westerlies in the equatorial western Pacific. Anomalous westerlies in the equatorial western Pacific are closely linked to the positive SST anomaly band in the subtropical northeastern Pacific, which persists until summer and develops toward the equator where it forces the overlying atmosphere,

strengthening anomalous westerlies in the equatorial western Pacific.

To further elucidate the connection between the equatorial eastern Pacific warming and SST anomalies in the subtropical northeastern Pacific, we computed the lagged partial regressions of the JAS(0) SST and 850 hPa winds anomalies onto the JFM(0) NPO index after removing the influences of AMJ(0) SST anomalies in the subtropical northeastern Pacific (see Figure C3 below). With the influences of SST anomalies in the subtropical northeastern Pacific removed, the equatorial eastern Pacific warming disappears during JAS(0), confirming that the equatorial eastern Pacific warming may be related to SST anomalies in the subtropical northeastern Pacific.

Figure C3. Lagged partial regressions of the JAS(0) SST and 850 hPa winds anomalies onto the JFM(0) NPO index after removing the influences of the AMJ(0) SST anomalies in the subtropical northeastern Pacific (15°–25°N and 150°–120°W). Only SST and 850 hPa wind anomalies significant at the 95% confidence level are shown.

8. *I suggest adding surface winds in all the figures showing SLP anomalies in Supplementary Fig. 11 to see if the circulation anomalies associated with the negative SLP anomalies around Hawaii weakened the trade winds in the equatorial Pacific.*

Response: Thanks for the suggestion from the reviewer. Following the reviewer's

suggestion, we have added surface winds in all figures showing SLP anomalies in former Supplementary Figure 11 (please see Supplementary Figure 9 of the revised manuscript).

9. *In this study the Pacific meridional mode (PMM) is mentioned. I suggest using the North Pacific meridional mode (NPMM) instead of PMM in the manuscript because in the South Pacific Ocean there exists another PMM, the South Pacific meridional mode (SPMM), which is also closely related with ENSO (Min et al., 2017, J. Climate, 30, 1705–1720).*

Response: Thanks for the constructive suggestion from the reviewer. Following the reviewer's suggestion, we have replaced the Pacific meridional mode (PMM) with the North Pacific meridional mode (NPMM) in the revised manuscript. The references related to the SPMM (e.g., Zhang et al. 2014; Min et al. 2017) have been added in the revised manuscript.

Zhang, H., Clement, A., & Di Nezio, P. The South Pacific meridional mode: A mechanism for ENSO-like variability. *J. Clim.* 27, 769–783 (2014).

Min, Q. Y., Su, J. Z., & Zhang, R. H. Impact of the South and North Pacific meridional modes on the El Niño–Southern oscillation: Observational analysis and comparison. *J. Clim.* 30, 1705–1720 (2017).

10. *The historical simulations of CMIP5/6 models were from 1900 to 1999, during which the three multi-year El Niño events of 1957/58/59, 1968/69/70, and 1986/87/88 were included. I suggest the authors compare the three events in simulations with those in observations to see how well the simulated ones in representing the observed ones.*

Response: We appreciate the suggestion from the reviewer. We would like to clarify that simulated multi-year El Niño events in CMIP5/6 models do not necessarily match the observed ones because the historical simulations of CMIP5/6 models are based on continuous uninitialized simulations. Therefore, it seems not appropriate to compare the simulated and observed individual

multi-year El Niño events. However, we can compare the composited evolutions of multi-year El Niño events in CMIP5/6 models and observations. Our results have shown that the CMIP5/6 models can reasonably well reproduce the composited evolutions of the NPO-preceded multi-year El Niño events, namely, that the NPO-preceded El Niño events are characterized by a pattern of SST anomalies that resemble that of the CP El Niño during JFM(1), which often re-intensifies in the subsequent winter and finally develops into a multi-year El Niño event (please see Supplementary Figure 15 of the revised manuscript).

The present study focuses on establishing the evidence that NPO dynamics act as a driver for multi-year El Niño events. The historical simulations of CMIP5/6 models provide additional evidence for the dynamic link between the NPO and multi-year El Niño and thus support the main findings from the observation data. A more detailed comparison of multi-year El Niño events in CMIP5/6 models and observations is not the focus of the present study. We fully agree with the reviewer that it is very necessary to evaluate the performance of CMIP5/6 models in simulating multi-year El Niño events. In the future, we will conduct a detailed and in-depth assessment of the CMIP5/6 model's performance in simulating multi-year ENSO events as suggested by the reviewer.

11. *In this study the role played by NPO in multi-year El Niño events were stressed. In fact, the effect from the subtropical southern Pacific should also take effect in the multiyear El Niño events. For example, the evolution of the long-lasting El Niño events in 2014/15/16 were affected by the system in the subtropical southern Pacific (Min et al., 2015, Geophys. Res. Lett., 42, 6762–6770; Su et al., 2018, J. Climate, 31, 877–893). I suggest adding necessary discussions in the manuscript by referring these previous studies.*

Response: We fully agree with the reviewer that forcing from the subtropical South Pacific could also have an impact on the evolutions of multi-year El Niño events. Following the reviewer's suggestion, we have added the following

sentence in the section “Summary and discussion” of the revised manuscript (please see lines 371-373):

“For example, some studies have suggested that forcing from the southeastern subtropical Pacific could also have an impact on the evolutions of the 2014/15/16 El Niño event (Min et al. 2015; Su et al. 2018).”

Min, Q. Y., Su, J. Z., Zhang, R. H., & Rong, X. Y. What hindered the El Niño pattern in 2014? *Geophys. Res. Lett.* **42**, 6762–6770 (2015).

Su, J. Z., Zhang, R. H., Rong, X. Y., Min, Q. Y., & Zhu, C. W. Sea surface temperature in the subtropical Pacific boosted the 2015 El Niño and hindered the 2016 La Niña. *J. Clim.* **31**, 877–893 (2018).

Reviewer #1 (Remarks to the Author):

The authors have satisfactorily addressed my concerns. I thus recommend publication.

Reviewer #2 (Remarks to the Author):

The authors have addressed all my concerns. I recommend the manuscript to be accepted for publication.

Reviewer #3 (Remarks to the Author):

The authors have addressed my concerns reasonably. The manuscript is much improved. I suggest accepting this manuscript for publication in its current form.

**"Multi-year El Niño events tied to the North Pacific Oscillation"
(NCOMMS-21-46528A) by Ding et al.
Responses to Reviewers #1, #2 and #3**

June 6, 2022

REVIEWERS' COMMENTS

Reviewer #1 (Remarks to the Author):

The authors have satisfactorily addressed my concerns. I thus recommend publication.

Reviewer #2 (Remarks to the Author):

The authors have addressed all my concerns. I recommend the manuscript to be accepted for publication.

Reviewer #3 (Remarks to the Author):

The authors have addressed my concerns reasonably. The manuscript is much improved. I suggest accepting this manuscript for publication in its current form.

Response: We would like to thank the reviewers for their valuable time and effort in reviewing our manuscript. We sincerely appreciate all helpful comments and suggestions raised by the reviewers, which help us to improve the quality of our manuscripts.